# LUV-Net: Multi-Pattern Lung Ultrasound Video Classification through Pattern-Specific Attention with Efficient Temporal Feature Extraction

**Jung Hoon Lee**[1,2,3]                                       CROP2292@SNU.AC.KR
**Changi Kim** [1,2,3]                                         FRZ2YROOM@SNU.AC.KR
**Jinwoo Lee**[4]                                             REALRAIN7@SNU.AC.KR
**Si Mong Yoon**[4]                                           DOSTARK1986@GMAIL.COM
**Kyung-Eui Lee**[4]                                          SYKU1@HANMAIL.NET
**Hyun-Jun Park**[4]                                          PARKHJSNUH@GMAIL.COM
**Kwonhyung Hyung**[4]                                        HEROHKH@NAVER.COM
**Chang Min Park**[1,2,3]                                     MORPHIUS@SNU.AC.KR

[1] *Department of Interdisciplinary Program in Bioengineering, Seoul National University.*

[2] *Department of Radiology, Seoul National University Hospital, Seoul National University College of Medicine.*

[3] *Integrated Major in Innovative Medical Science, Seoul National University Graduate School.*

[4] *Division of Pulmonary and Critical Care Medicine, Department of Internal Medicine, Seoul National University College of Medicine.*

**Editors:** Accepted for publication at MIDL 2025

## Abstract

Lung ultrasound (LUS) has emerged as a crucial bedside imaging tool for critical care, yet its interpretation remains challenging due to its artifact-based nature and high operator dependency. While deep learning approaches offer promising solutions for LUS pattern analysis, existing methods are limited by their focus on single-pattern recognition or disease-specific classification, and insufficient modeling of temporal relationships in LUS video data. We propose LUV-Net (Lung Ultrasound Video Network), a novel deep learning model for multi-label classification of LUS patterns, combining pattern-specific attention mechanisms with temporal feature extraction. Our approach consists of two key modules: a spatial feature extraction module utilizing independent pattern-specific attention mechanisms, and a temporal feature extraction module designed to capture sequential relationships between adjacent frames. The model was evaluated using two distinct datasets: a development set of 341 LUS videos and a temporally separated validation set of 56 videos. Through 5-fold cross-validation, LUV-Net demonstrated superior performance in identifying all four LUS patterns (A-lines, B-lines, consolidation, and pleural effusion) compared to CNN-based and Transformer-based video models, achieving higher AUC scores across patterns. The model's interpretability was validated through visualization of pattern-specific attention regions, providing insights into its decision-making process. The code is publicly available at $https://github.com/iamhxxn2/LungUS_Video$.

**Keywords:** Video Multi-label Classification, Lung Ultrasound, Pattern-Specific Attention, Efficient Temporal Feature

## 1. Introduction

Point-of-care ultrasound (POCUS) has progressively proven its significance as a useful bedside imaging modality, crucial for the assessment of critically ill patients and facilitating both diagnostic and therapeutic decision-making processes (Zieleskiewicz et al., 2021; Shrestha et al., 2018). Lung ultrasound (LUS) has been shown to have higher sensitivity for pneumothorax and pleural effusion than chest radiography (CXR) (Shrestha et al., 2018; Brogi et al., 2017), offering advantages of being non-invasive, cost-effective, and portable. Therefore, LUS has considerable potential as an important tool in low- and middle-income countries (LMICs) (Marini et al., 2021; Shrestha et al., 2018; Buonsenso and De Rose, 2022). However, LUS interpretation presents significant challenges due to its artifact-based nature rather than direct lung anatomy visualization, making it highly operator-dependent. Additionally, the lack of qualified ultrasound professionals and insufficient training programs are significant obstacles to the application of LUS in clinical practice (Marini et al., 2021; Nhat et al., 2023; Lim et al., 2017).

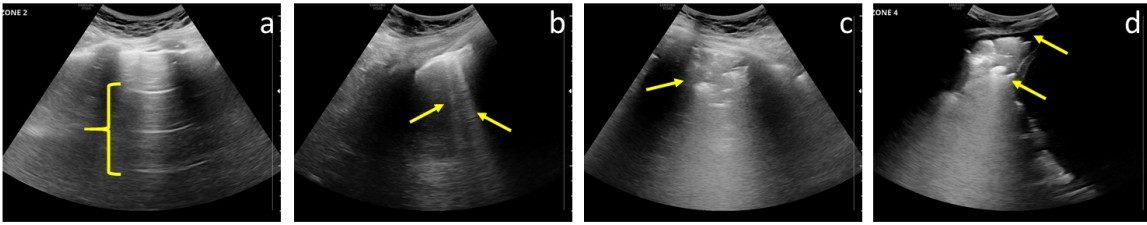

Figure 1: Example lung ultrasound frames and features: (a) A-lines, (b) B-line, (c) Consolidation, (d) Pleural effusion with consolidation

For LUS, there are several main patterns in lung ultrasound images, including A-line, B-line, consolidation, and pleural effusion, with healthy lungs typically exhibiting only A-lines while other patterns may emerge or coexist associated with different lung diseases(Ni et al., 2024), as shown in the examples in Figure 1. This characteristic inherently makes LUS pattern recognition a multi-label classification problem. However, recent research has focused mainly on the recognition of a single pattern (B-line) (Kerdegari et al., 2021; Arntfield et al., 2021) or, on multi-class classification for specific lung diseases(Nhat et al., 2023; Shea et al., 2023; Howell et al., 2024; Diaz-Escobar et al., 2021; Roy et al., 2020).

While deep learning methods have shown promise in automated video analysis (Kerdegari et al., 2021; Shea et al., 2023; De Rosa et al., 2022; Barros et al., 2021), applying video recognition techniques to LUS faces several challenges due to the fundamental differences between ultrasound and natural imagery. Current approaches primarily employ CNN+LSTM architecture or 3D convolution-based architectures (e.g., C3D(Tran et al., 2015), R2Plus1D(Tran et al., 2018)) to capture spatiotemporal features in LUS sequences (Shea et al., 2023; Barros et al., 2021; Dastider et al., 2021; Liu et al., 2024; Ebadi et al., 2021). These approaches focus on learning temporal dependencies across the entire video sequence. Smith, D. H. et al. (Smith et al., 2023) challenge this methodology, arguing that models developed for human action recognition are not optimal in some practical scenarios involving medical ultrasound and that models assuming temporal independence demonstrate better sample efficiency. In the specific case of LUS data, we hypothesize that

a hybrid approach considering both local temporal dependencies (relationships between a target frame and its neighboring frames) and frame-wise features might be more effective for accurate pattern recognition in LUS video.

**Contributions**: We introduce the Lung Ultrasound Video Network (LUV-Net), a deep learning model for multi-pattern recognition in LUS videos that combines pattern-specific attention with efficient temporal feature extraction.

## 2. Methods

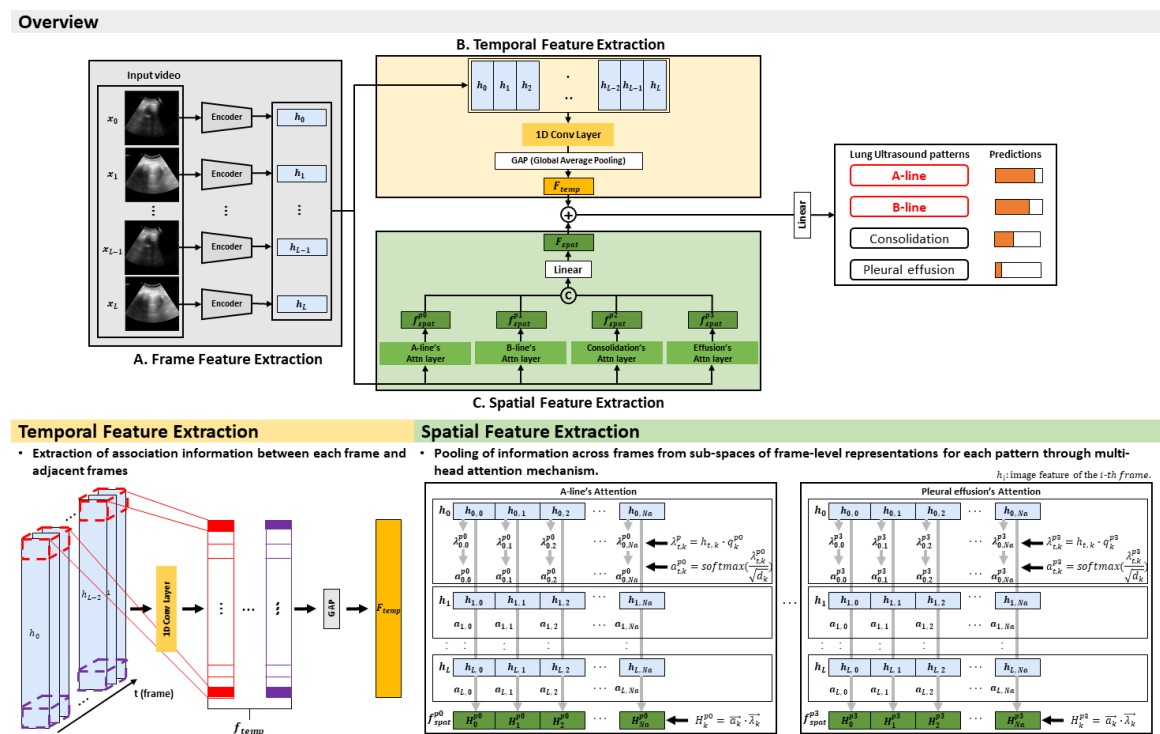

Figure 2: Proposed framework of the Lung Ultrasound Video Network (LUV-Net).

The proposed Lung Ultrasound Video Network (LUV-Net), as shown in Figure 2, consists of four main components: frame feature extraction, temporal feature extraction, spatial feature extraction, and a feature fusion stage.

### 2.1. Frame Feature Extraction

The input clip $X$ to the model is a sequence of $L$ frames, $X = (x_1, x_2, \ldots, x_L)$. Each frame is fed into the CNN encoder individually, embedding the features into $D$-dimensional vectors. The features for each frame are denoted as $h_1, h_2, \ldots, h_L$, where $h \in \mathbb{R}^D$. These frame-level features are then used in both the temporal feature extraction and spatial feature extraction networks.

## 2.2. Temporal Feature Extraction

Our temporal feature extraction module is designed to capture the dynamic relationships between consecutive frames in LUS video sequences. This process utilizes a 1D convolution operation to analyze temporal patterns across the video timeline. For a sequence of frame features $h_1, h_2, \ldots, h_L$, we apply a convolution kernel with size $k$ that processes overlapping windows of consecutive frames. This operation generates temporally aware features:

$$f_{temp} = Conv1D(h_1, h_2, \ldots, h_L) \tag{1}$$

where $f_{temp} \in \mathbb{R}^{L \times D}$ represents the extracted temporal features. The kernel size $k$ determines how many adjacent frames are analyzed together, while the stride controls the step size between windows. To obtain the final video-level temporal features, we aggregate the temporal features using Global Average Pooling (GAP), which summarizes the temporal information into a single feature vector:

$$F_{temp} = GAP(f_{temp}) \tag{2}$$

## 2.3. Spatial Feature Extraction

Our approach employs a pattern-specific spatial attention mechanism that builds upon the work of (Smith et al., 2023), with a key modification to handle different LUS patterns independently. While prior study applied attention mechanisms uniformly across all features, we recognize that different LUS patterns may require attention to different frames within the video sequence. Therefore, we apply separate attention mechanisms for each pattern.

Each frame representation, denoted as $h_1, h_2, \ldots, h_L$, is divided into $N_a$ segments for the attention heads, resulting in sub-representations $h_{t,k}$, where $t$ and $k$ index the frame and attention head, respectively. Each segment $h_{t,k} \in \mathbb{R}^{d_a}$ has dimensionality $d_a = D/N_a$. Following (Smith et al., 2023), we compute attention scores using learned global query vectors instead of comparing features across frames. The use of global query vectors is motivated by the inductive prior that the recognition task involves locating key pieces of information at any point in the sequence, with the learned queries acting as pattern-specific detectors. The attention mechanism computes dot products between frame representations and global query vectors: $\lambda_{t,k} = h_{t,k} \cdot q_k^p$, where superscripts p0, p1, p2, and p3 correspond to A-line, B-line, consolidation, and pleural effusion patterns respectively. These scores are scaled by $\sqrt{d_k}$ for numerical stability and normalized across the temporal dimension using the softmax function to obtain attention weights: $a_{t,k}^p = \exp\left(\lambda_{t,k}^p / \sqrt{d_k}\right)$. These scores are normalized using the softmax function to obtain attention vectors $a_{t,k}$ that is the attention vector derived from the normalized scores. The video-level representation from each head is computed as the weighted sum of the frame representations: $H_k^p = \sum_{t=1}^{L} a_{t,k}^p \cdot h_{t,k}$. For each pattern (p), we concatenate the outputs from all attention heads to obtain the pattern-specific spatial feature representation:

$$f_{spat}^p = concat([H_1^p, H_2^p, \ldots, H_{N_a}^p]) \tag{3}$$

Finally, we combine the pattern-specific features and project them through a linear layer to match the temporal feature dimensionality:

$$F_{spat} = Linear(concat(f_{spat}^{p0}, f_{spat}^{p1}, f_{spat}^{p2}, f_{spat}^{p3})) \tag{4}$$

## 2.4. Feature Fusion Stage

In the Feature Fusion Stage, the temporal and spatial features are combined to create a comprehensive video representation. This is achieved by summing the temporal feature representation $F_{temp}$ with the spatial feature representation $F_{spat}$. Subsequently, the video-level prediction can then be computed using a fully connected linear layer for multi-label classification of LUS patterns.

## 3. Materials and Experimental Setup

### 3.1. Datasets

**Data Collection and Annotation.** The LUS scans were performed using a HS60 Ultrasound Machine (Samsung Healthcare, Republic of Korea) with a low–medium frequency (3–5 MHz) convex probe. Following the protocol recommended by the Bedside Lung Ultrasound in Emergency (BLUE) protocol (Lichtenstein and Meziere, 2008), three lung points—anterior, lateral, and posterior areas—were assessed by the operators, ensuring that a minimum of six videos were acquired per patient. From January to December 2023, we collected 370 LUS videos, each with a duration of approximately 5 seconds, from 36 patients in the ICU of Seoul National University Hospital(SNUH) for the model development set. Videos with poor quality due to blurring or darkness, which hindered the differentiation of LUS patterns, were excluded from the dataset. As a result, the final dataset used for model training consisted of 341 LUS videos from 35 patients. Additionally, for model validation, a temporally separated test set was collected from 11 patients in the SNUH ICU between January and December 2024, resulting in the acquisition of 56 LUS videos. Each video ranged from 5 to 8 seconds in duration, with a frame rate of 30 frames per second (fps) and a resolution of $924 \times 1232$ pixels. In the development set, each video was independently labeled by two clinicians with over 1 year of experience in lung ultrasound (LUS), who annotated the regions containing LUS patterns at the frame level. In cases where there was agreement between the two clinicians, their consensus label was adopted as the final label. For disagreements, a clinician with over 8 years of experience conducted the final review. For the temporally separated test set, all labeling was performed by a single clinician with over eight years of experience in lung ultrasound.

**Preprocessing Stage.** For training the model, we conducted several preprocessing steps on the video data. First, we segmented each video into one-second clips (30 frames) with a 20% frame overlap (6 frames) between consecutive frames. This preprocessing resulted in 2,588 clips from 370 videos for the model development set, and 366 clips from 56 videos for the temporally separated test set. Additionally, we downsampled all video frames to a uniform resolution of $256 \times 256$ pixels. To enhance data diversity, we implemented augmentation techniques including random horizontal flips and controlled rotations up to 10 degrees. The augmented images were subsequently converted into tensor format for deep learning processing. To ensure robust evaluation, we randomly selected 10% of the total data as a held-out test set and subjected the remaining 90% to 5-fold cross-validation, ensuring strict separation of patient data across all sets.

### 3.2. Implementation Details

In this study, we compared the performance of our proposed LUV-Net model with CNN-based video models (USVN (Smith et al., 2023), C3D, R2Plus1D, and CNN+LSTM) and Transformer-based video models (MViT-B(Fan et al., 2021) and Swin-B(Liu et al., 2022)). Additionally, we evaluated a frame-based method that processes frame-level features independently and derives final video predictions by pooling across the frames. To ensure consistency in feature extraction, LUV-Net, USVN, and CNN+LSTM models employed ImageNet pre-trained DenseNet-161(Huang et al., 2017) as the default image encoder. The CNN+LSTM model utilized a single LSTM layer for sequential feature processing. Furthermore, to explore the impact of using a lighter backbone, we conducted an additional experiment replacing DenseNet-161 with ResNet-50(He et al., 2016) as the encoder in LUV-Net, analyzing its effect on model efficiency and performance. For a fair comparative analysis with models that do not use pretrained weights (C3D, R2Plus1D, MViT-B, and Swin-B), we also evaluated LUV-Net with a non-pretrained DenseNet-161 backbone. All baseline models were trained on our dataset under identical experimental conditions to ensure fair comparison.

For LUV-Net, we optimized the key hyperparameters of the temporal and spatial feature extraction networks through a systematic hyperparameter search (detailed in Table 8, Table 9, Appendix C). Through this process, we determined that a 1D convolutional kernel size of 13 achieved the best validation performance across all evaluation metrics. The kernel size selection was based on an empirical analysis of different sizes (ranging from 1 to 29), where kernel size 13 consistently provided optimal results in both macro-AUC and micro-AUC scores (as detailed in Table 8). This suggests that a receptive field spanning 13 consecutive frames effectively captures the local temporal dependencies in lung ultrasound videos, balancing sensitivity and specificity in LUS pattern recognition. Similarly, for the spatial feature extraction network, we selected 8 attention heads ($N_a = 8$) based on an extensive evaluation of different head configurations (ranging from 1 to 96). The results indicated that using 8 attention heads maximized the model's performance without introducing excessive computational overhead, as shown in Table 9. Increasing the number of attention heads beyond this point did not yield further improvements, confirming that this configuration effectively balances feature representation across different LUS patterns.

During training, we used the Adam optimizer(Kingma and Ba, 2014) with a learning rate of 1e-6, a batch size of 4, and trained the model for up to 150 epochs. To prevent overfitting, we implemented an early stopping mechanism with a patience value of 50, ensuring that the best-performing model was selected based on validation loss. The final thresholds for classification were determined at the epoch with the lowest validation loss, and these were applied consistently to both the development set and the temporally separated validation set. All training experiments were conducted using a single NVIDIA A100 GPU.

## 4. Results

The evaluation was conducted using 5-fold cross-validation on both development and temporally separated sets, with an additional comparative analysis between LUV-Net and its variant without temporal feature extraction. The main tables present the mean and standard deviation values for each metric, with bold and underlined values indicating the best

and second-best performances, respectively. The detailed results for each fold of the 5-fold cross-validation, including P values ($p < 0.05$) and 95% confidence intervals (CI), are presented in Tables 5, 6, and 7 in Appendix A.

## 4.1. Development Dataset

Table 1 shows the 5-fold cross-validation results for the development set. Our proposed LUV-Net demonstrates superior performance across most metrics, achieving the highest scores in B-line (AUC: 0.834±0.014), consolidation detection (AUC: 0.853±0.019), and overall performance (Micro: 0.888±0.009, Macro: 0.894±0.009). While the frame-based method shows competitive performance in A-line (AUC: 0.926±0.007) and pleural effusion detection (AUC: 0.973±0.004), LUV-Net maintains more consistent performance across all patterns. USVN achieves competitive results in pleural effusion (AUC: 0.931±0.037) and consolidation (AUC: 0.838±0.010), but shows high variance in B-line detection (0.770±0.137). C3D, R2Plus1D and CNN+LSTM show notably lower performance, particularly in A-line and B-line detection. Transformer-based video models, MViT-B and Swin-B, show significantly lower performance compared to CNN-based approaches. MViT-B achieves an overall Micro AUC of 0.619±0.055, while Swin-B obtains 0.601±0.064. For the additional baselines, LUV-Net with ResNet-50 exhibits comparable performance across patterns, achieving a Micro AUC of 0.808±0.030 and a Macro AUC of 0.806±0.034. Notably, despite lower performance than its LUV-Net (DenseNet-161)$^{\dagger}$, LUV-Net (DenseNet-161) still consistently outperforms non-pretrained baselines (C3D, R2Plus1D, MViT-B, and Swin-B), demonstrating the inherent effectiveness of our architectural design independent of transfer learning advantages.

| Input Type | Model | AUC | | | | Avg | |
| --- | --- | --- | --- | --- | --- | --- | --- |
| | | A-line | B-line | Consolidation | Pleural effusion | Micro | Macro |
| Frame (Image) | frame-based | **0.926±0.007** | 0.800±0.033 | 0.831±0.018 | **0.973±0.004** | 0.881±0.008 | 0.883±0.011 |
| Video | MViT-B | 0.598±0.085 | 0.527±0.130 | 0.620±0.066 | 0.752±0.109 | 0.619±0.055 | 0.626±0.066 |
| | Swin-B | 0.582±0.095 | 0.574±0.130 | 0.604±0.077 | 0.714±0.100 | 0.601±0.064 | 0.620±0.074 |
| | C3D | 0.789±0.062 | 0.626±0.038 | 0.788±0.023 | 0.930±0.002 | 0.791±0.025 | 0.792±0.023 |
| | R2Plus1D | 0.726±0.051 | 0.619±0.078 | 0.801±0.031 | 0.830±0.071 | 0.720±0.031 | 0.746±0.035 |
| | CNN(Densenet-161)$^{\dagger}$+LSTM | 0.420±0.066 | 0.353±0.032 | 0.737±0.043 | 0.911±0.042 | 0.587±0.023 | 0.607±0.007 |
| | USVN(Densenet-161)$^{\dagger}$ | 0.879±0.080 | 0.770±0.137 | **0.879±0.080** | 0.931±0.037 | 0.846±0.062 | 0.856±0.058 |
| | LUV-Net(Resnet-50)$^{\dagger}$ | 0.867±0.026 | 0.778±0.020 | 0.762±0.025 | 0.806±0.140 | 0.808±0.030 | 0.806±0.034 |
| | LUV-Net(Densenet-161)$^{\dagger}$ | 0.918±0.013 | **0.834±0.014** | 0.853±0.019 | 0.966±0.010 | **0.888±0.009** | **0.894±0.009** |
| | LUV-Net(Densenet-161) | 0.865±0.026 | 0.644±0.059 | 0.872±0.033 | 0.849±0.098 | 0.809±0.045 | 0.809±0.034 |

Table 1: Results on the development set. $^{\dagger}$ indicates ImageNet pretrained model.

## 4.2. Temporally Separated Dataset

On the temporally separated dataset (Table 2), LUV-Net(DenseNet-161) maintains robust performance with the highest AUC in A-line detection (0.835±0.057) and overall metrics (Micro:0.858±0.023, Macro:0.844±0.015). While USVN achieves the highest performance in consolidation detection (0.846±0.022) and shows comparable B-line detection (0.848±0.039), LUV-Net demonstrates superior performance in other patterns, particularly in A-line and pleural effusion detection. C3D shows competitive performance in pleural effusion detection (0.882±0.032), but our model demonstrates more balanced performance across all patterns, confirming its effectiveness in multi-label LUS pattern classification. For LUV-Net(ResNet-50), performance remains competitive, with a Micro AUC of 0.836±0.017 and a Macro AUC of 0.815±0.013, though slightly lower than its DenseNet-161 counterpart. MViT-B and Swin-B show lower performance, with Micro AUC of 0.740±0.093 and

0.737±0.047, respectively. Similar to the development set, while the LUV-Net (DenseNet-161) did not reach the performance of LUV-Net (DenseNet-161)$^\dagger$, it outperformed other baseline models, excelling in A-line (AUC: 0.841±0.030) and pleural effusion detection (AUC: 0.885±0.041).

| Input Type | Model | AUC | | | | Avg | |
|---|---|---|---|---|---|---|---|
| | | A-line | B-line | Consolidation | Pleural effusion | Micro | Macro |
| Frame (Image) | frame-based | 0.763±0.025 | **0.893±0.005** | 0.772±0.019 | 0.812±0.023 | 0.809±0.031 | 0.813±0.014 |
| Video | MViT-B | 0.633±0.118 | 0.538±0.043 | 0.633±0.047 | 0.825±0.100 | 0.740±0.093 | 0.659±0.064 |
| | Swin-B | 0.642±0.104 | 0.546±0.044 | 0.632±0.083 | 0.751±0.096 | 0.737±0.047 | 0.645±0.071 |
| | C3D | 0.795±0.030 | 0.703±0.047 | 0.747±0.010 | 0.882±0.032 | 0.848±0.025 | 0.784±0.025 |
| | R2Plus1D | 0.659±0.056 | 0.634±0.010 | 0.684±0.027 | 0.868±0.011 | 0.734±0.030 | 0.713±0.014 |
| | CNN(Densenet-161)$^\dagger$+LSTM | 0.366±0.092 | 0.491±0.047 | 0.756±0.042 | 0.754±0.084 | 0.602±0.069 | 0.593±0.029 |
| | USVN(Densenet-161)$^\dagger$ | 0.795±0.024 | 0.848±0.039 | **0.846±0.022** | 0.845±0.068 | 0.824±0.049 | 0.833±0.027 |
| | LUV-Net(Resnet-50)$^\dagger$ | 0.790±0.032 | 0.764±0.011 | 0.824±0.008 | 0.877±0.035 | 0.836±0.017 | 0.815±0.013 |
| | LUV-Net(Densenet-161)$^\dagger$ | 0.835±0.057 | 0.862±0.022 | 0.799±0.021 | 0.873±0.026 | **0.858±0.023** | **0.844±0.015** |
| | LUV-Net(Densenet-161) | **0.841±0.030** | 0.778±0.035 | 0.744±0.019 | **0.885±0.041** | 0.795±0.048 | 0.813±0.019 |

Table 2: Results on the temporally separated set. $^\dagger$ indicates ImageNet pretrained model.

### 4.3. Effectiveness of Temporal Feature Extraction

Table 3 compares the performance of LUV-Net(DenseNet-161) with and without temporal feature extraction on the development set. Incorporating temporal features consistently improved performance across all patterns, achieving higher Macro AUC (0.894±0.009 vs 0.885±0.011) and Micro AUC (0.888±0.009 vs 0.880±0.014), which demonstrates the effectiveness of temporal feature extraction in enhancing pattern recognition.

| | AUC | | | | Avg | |
|---|---|---|---|---|---|---|
| | A-line | B-line | Consolidation | Pleural effusion | Micro | Macro |
| LUV-Net (w/o temporal) | 0.910±0.017 | 0.826±0.019 | 0.841±0.018 | 0.956±0.009 | 0.880±0.014 | 0.885±0.011 |
| LUV-Net (w/ temporal) | **0.918±0.013** | **0.834±0.015** | **0.853±0.019** | **0.966±0.010** | **0.888±0.009** | **0.894±0.009** |

Table 3: Temporal feature extraction with / without on development set

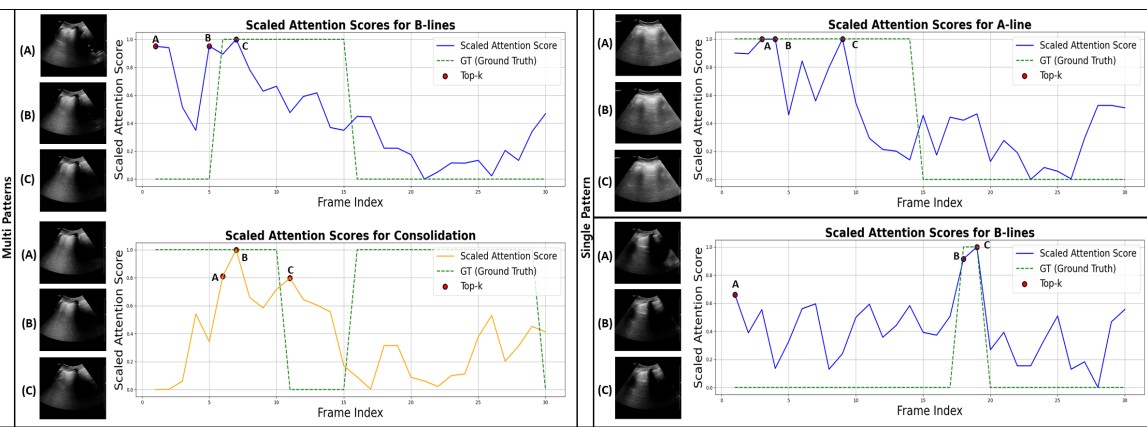

Figure 3: Visualization of attention scores and corresponding top-3 frames for each pattern.

### 4.4. Qualitative Analysis

To further investigate the interpretability of the proposed LUV-Net(DenseNet-161)$^{†}$ model, we conducted a qualitative analysis by visualizing the attention scores across video frames for multiple patterns and extracting the top-$k$ frames with the highest scores, as detailed in Appendix B. Figure 3 shows examples from the development set, including both multi-patterns and single-pattern clips (additional examples from the temporally separated set can be found in Appendix B). The top-3 frames for each pattern are marked with red dots, with (A), (B), and (C) denoting these frames. The green dashed lines represent the ground truth, where 1 indicates presence and 0 indicates absence of the pattern. Our model identifies critical frames and attends to the relevant regions of each pattern.

### 4.5. Computational Efficiency Analysis

Table 4 compares model parameters, GFLOPs, and inference time per video across different architectures. LUV-Net (DenseNet-161) achieves the highest classification performance but at a higher computational cost (135.99M parameters, 304.68 GFLOPs, 0.065s per video). In contrast, LUV-Net (ResNet-50) offers a more efficient alternative (163.57 GFLOPs, 0.027s) while maintaining competitive performance. Transformer-based models (e.g., MViT-B, Swin-B) had fewer parameters but exhibited longer inference times due to self-attention overhead. USVN and CNN+LSTM showed similar computational loads to LUV-Net but lower accuracy, and C3D-based models (C3D, R2Plus1D) required the highest GFLOPs with limited efficiency gains.

| Model / Metric | LUV_Net(DenseNet-161) | LUV_Net(ResNet-50) | MViT-B | Swin-B | USVN(DenseNet-161) | C3D | R2Plus1D | CNN(DenseNet-161)+LSTM |
|---|---|---|---|---|---|---|---|---|
| Param | 135.99 | 121.03 | 36.00 | 58.72 | 26.26 | 214.33 | 63.51 | 28.54 |
| GFLOPs | 304.68 | 163.57 | 137.91 | 200.65 | 302.74 | 369.14 | 764.45 | 302.81 |
| Inference Time(s) | 0.065 | 0.027 | 0.050 | 0.121 | 0.062 | 0.026 | 0.094 | 0.062 |

Table 4: Comparison of model parameters and computational cost

## 5. Discussion and Conclusions

We propose LUV-Net, a deep learning model for multi-label classification of LUS patterns in ultrasound videos. Our 5-fold cross-validation demonstrates superior performance across all four LUS patterns compared to baseline models. By integrating spatial attention mechanisms and temporal feature extraction, LUV-Net enhances both classification performance and interpretability, effectively identifying when and where specific patterns appear. Despite their efficacy in natural video domains, CNN-based and Transformer-based baseline models underperformed on LUS data, likely stemming from fundamental differences between natural and ultrasound videos. While models for human action recognition primarily rely on temporal sequence modeling, the analysis of lung ultrasound (LUS) data is better addressed by approaches that emphasize frame-level spatial features, which are more diagnostically informative in the context of medical ultrasound imaging. However, Limitations of our study include the single-institution source of our dataset, potentially restricting generalizability. Additionally, our spatial and temporal modules could be further optimized for real-time clinical applications through model compression techniques. Real-world validation through reader studies with practicing clinicians remains essential to assess LUV-Net's practical clinical utility.

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

# Appendix A. 5-Fold Cross Validation Results

## A.1. Results on the development set

| Fold # | Input Type | Model | AUC | | | | Avg | |
|---|---|---|---|---|---|---|---|---|
| | | | A-line | B-line | Consolidation | Pleural effusion | Micro | Macro |
| Fold 0 | Frame (Image) | frame-based | **0.917 (0.886-0.949)** | 0.837 (0.800-0.875) | 0.821 (0.775-0.866) | **0.979 (0.965-0.994)** | 0.885 | 0.889 |
| | Video | MViT-B | 0.545 (0.474-0.616)* | 0.615 (0.557-0.673)* | 0.563 (0.506-0.621)* | 0.673 (0.587-0.759)* | 0.570 | 0.601 |
| | | Swin-B | 0.564 (0.495-0.634)* | 0.598 (0.543-0.653)* | 0.608 (0.553-0.663)* | 0.719 (0.644-0.794)* | 0.592 | 0.624 |
| | | C3D | 0.758 (0.708-0.808)* | 0.640 (0.586-0.694)* | 0.778 (0.733-0.822)* | 0.931 (0.902-0.960)* | 0.801 | 0.811 |
| | | R2Plus1D | 0.707 (0.657-0.757)* | 0.573 (0.513-0.632)* | 0.802 (0.749-0.854) | 0.911 (0.878-0.945)* | 0.703 | 0.750 |
| | | CNN(Densenet-161)†+LSTM | 0.368 (0.315-0.421)* | 0.347 (0.288-0.406)* | 0.760 (0.705-0.815)* | 0.924 (0.891-0.957)* | 0.602 | 0.601 |
| | | USVN(Densenet-161)† | 0.902 (0.865-0.939) | 0.834 (0.796-0.872) | 0.829 (0.788-0.871)* | 0.948 (0.926-0.969)* | 0.868 | 0.880 |
| | | LUV-Net(Resnet-50)† | 0.852 (0.0.813-0.890)* | 0.772 (0.724-0.819)* | 0.773 (0.721-0.825)* | 0.977 (0.964-0.989) | 0.853 | 0.845 |
| | | LUV-Net(Densenet-161)† | 0.900 (0.860-0.940) | **0.841 (0.802-0.880)** | 0.851 (0.810-0.892) | 0.974 (0.959-0.989) | **0.880** | **0.893** |
| | | LUV-Net(Densenet-161) | 0.831 (0.789-0.873)* | 0.669 (0.615-0.723)* | **0.880 (0.848-0.913)** | 0.943 (0.909-0.977)* | 0.842 | 0.833 |
| Fold 1 | Frame (Image) | frame-based | **0.924 (0.897-0.951)*** | 0.791 (0.747-0.832) | 0.833 (0.783-0.883)* | **0.971 (0.958-0.984)** | 0.875 | 0.880 |
| | Video | MViT-B | 0.669 (0.607-0.730)* | 0.419 (0.360-0.478)* | 0.712 (0.662-0.761)* | 0.834 (0.790-0.877)* | 0.656 | 0.660 |
| | | Swin-B | 0.748 (0.698-0.798)* | 0.499 (0.440-0.558)* | 0.746 (0.697-0.796)* | 0.895 (0.851-0.939)* | 0.720 | 0.724 |
| | | C3D | 0.762 (0.712-0.811)* | 0.564 (0.509-0.618)* | 0.814 (0.774-0.855)* | 0.928 (0.899-0.956) | 0.775 | 0.769 |
| | | R2Plus1D | 0.653 (0.598-0.708)* | 0.534 (0.479-0.590)* | 0.805 (0.755-0.856)* | 0.814 (0.750-0.878)* | 0.686 | 0.703 |
| | | CNN(Densenet-161)†+LSTM | 0.478 (0.421-0.536)* | 0.328 (0.274-0.382)* | 0.724 (0.667-0.782)* | 0.936 (0.904-0.967)† | 0.605 | 0.618 |
| | | USVN(Densenet-161)† | 0.729 (0.676-0.783)* | 0.536 (0.479-0.594)* | 0.848 (0.808-0.888) | 0.866 (0.814-0.919)* | 0.734 | 0.747 |
| | | LUV-Net(Resnet-50)† | 0.827 (0.782-0.872)* | 0.756 (0.710-0.802)* | 0.775 (0.728-0.821)* | 0.636 (0.523-0.749)* | 0.784 | 0.755 |
| | | LUV-Net(Densenet-161)† | 0.914 (0.886-0.943) | **0.821 (0.779-0.862)** | 0.860 (0.822-0.897) | 0.950 (0.927-0.972) | **0.882** | **0.887** |
| | | LUV-Net(Densenet-161) | 0.877 (0.843-0.912)* | 0.696 (0.642-0.750)* | **0.892 (0.861-0.923)** | 0.871 (0.834-0.909)* | 0.826 | 0.836 |
| Fold 2 | Frame (Image) | frame-based | 0.930 (0.905-0.955) | 0.747 (0.695-0.798)* | 0.820 (0.774-0.866) | **0.971 (0.955-0.988)** | 0.876 | 0.868 |
| | Video | MViT-B | 0.506 (0.438-0.573)* | 0.682 (0.630-0.734)* | 0.560 (0.502-0.618)* | 0.632 (0.542-0.722)* | 0.611 | 0.597 |
| | | Swin-B | 0.573 (0.505-0.642)* | 0.695 (0.644-0.747)* | 0.536 (0.476-0.595)* | 0.643 (0.552-0.734)* | 0.564 | 0.614 |
| | | C3D | 0.714 (0.663-0.766)* | 0.648 (0.590-0.707)* | 0.756 (0.708-0.805)* | 0.929 (0.904-0.955)* | 0.756 | 0.764 |
| | | R2Plus1D | 0.737 (0.692-0.782)* | 0.732 (0.674-0.790)* | 0.747 (0.692-0.802)* | 0.828 (0.767-0.888)* | 0.743 | 0.763 |
| | | CNN(Densenet-161)†+LSTM | 0.350 (0.299-0.401)* | 0.349 (0.290-0.408)* | 0.779 (0.727-0.831)* | 0.944 (0.917-0.970)† | 0.585 | 0.607 |
| | | USVN(Densenet-161)† | 0.924 (0.896-0.951) | 0.821 (0.780-0.862) | 0.833 (0.792-0.874) | 0.953 (0.932-0.974) | 0.881 | 0.884 |
| | | LUV-Net(Resnet-50)† | 0.886 (0.851-0.921)* | 0.785 (0.738-0.833) | 0.725 (0.673-0.777)* | **0.966 (0.945-0.986)** | 0.819 | 0.842 |
| | | LUV-Net(Densenet-161)† | **0.933 (0.908-0.958)** | **0.827 (0.786-0.868)** | 0.834 (0.796-0.871) | 0.963 (0.944-0.983) | **0.890** | **0.890** |
| | | LUV-Net(Densenet-161) | 0.838 (0.796-0.880)* | 0.551 (0.490-0.612)* | **0.881 (0.847-0.914)*** | 0.743 (0.667-0.818)* | 0.725 | 0.755 |
| Fold 3 | Frame (Image) | frame-based | **0.934 (0.910-0.958)** | 0.823 (0.782-0.865) | 0.859 (0.818-0.899) | 0.969 (0.945-0.993) | 0.894 | 0.897 |
| | Video | MViT-B | 0.728 (0.673-0.782)* | 0.587 (0.522-0.652)* | 0.690 (0.641-0.739)* | 0.923 (0.897-0.950)* | 0.704 | 0.733 |
| | | Swin-B | 0.570 (0.501-0.639)* | 0.712 (0.659-0.765)* | 0.593 (0.538-0.649)* | 0.710 (0.632-0.787)* | 0.599 | 0.648 |
| | | C3D | 0.859 (0.822-0.896)* | 0.670 (0.615-0.725)* | 0.795 (0.753-0.836)* | 0.930 (0.902-0.957)* | 0.820 | 0.815 |
| | | R2Plus1D | 0.789 (0.746-0.832)* | 0.676 (0.616-0.736)* | 0.831 (0.786-0.876)* | 0.869 (0.819-0.919)* | 0.762 | 0.793 |
| | | CNN(Densenet-161)†+LSTM | 0.395 (0.340-0.450)* | 0.332 (0.276-0.388)* | 0.758 (0.703-0.813)* | 0.924 (0.891-0.957)* | 0.593 | 0.601 |
| | | USVN(Densenet-161)† | 0.924 (0.895-0.954) | 0.849 (0.813-0.886) | 0.852 (0.815-0.888)* | 0.951 (0.930-0.972)* | 0.884 | 0.895 |
| | | LUV-Net(Resnet-50)† | 0.900 (0.870-0.931) | 0.813 (0.771-0.856)* | 0.794 (0.746-0.843)* | 0.707 (0.607-0.806)* | 0.817 | 0.806 |
| | | LUV-Net(Densenet-161)† | 0.919 (0.891-0.948) | **0.855 (0.817-0.893)** | 0.882 (0.848-0.916) | **0.975 (0.958-0.992)** | **0.903** | **0.909** |
| | | LUV-Net(Densenet-161) | 0.881 (0.844-0.918) | 0.704 (0.650-0.757)* | 0.807 (0.764-0.851)* | 0.960 (0.942-0.978)* | 0.848 | 0.840 |
| Fold 4 | Frame (Image) | frame-based | 0.924 (0.897-0.952) | 0.803 (0.757-0.848) | 0.822 (0.779-0.864) | **0.977 (0.964-0.990)** | 0.877 | 0.882 |
| | Video | MViT-B | 0.542 (0.472-0.611)* | 0.333 (0.279-0.386)* | 0.577 (0.520-0.634)* | 0.697 (0.624-0.770)* | 0.554 | 0.539 |
| | | Swin-B | 0.453 (0.388-0.518)* | 0.364 (0.310-0.419)* | 0.537 (0.477-0.598)* | 0.604 (0.506-0.703)* | 0.530 | 0.492 |
| | | C3D | 0.853 (0.815-0.891)* | 0.606 (0.549-0.663)* | 0.798 (0.756-0.839)* | 0.934 (0.908-0.960)* | 0.805 | 0.799 |
| | | R2Plus1D | 0.745 (0.698-0.791)* | 0.581 (0.524-0.638)* | 0.819 (0.769-0.869) | 0.730 (0.675-0.786)* | 0.704 | 0.721 |
| | | CNN(Densenet-161)†+LSTM | 0.509 (0.447-0.570)* | 0.409 (0.347-0.472)* | 0.666 (0.605-0.726)* | 0.836 (0.779-0.892)* | 0.550 | 0.606 |
| | | USVN(Densenet-161)† | 0.916 (0.884-0.947) | 0.810 (0.767-0.853) | 0.830 (0.787-0.872) | 0.938 (0.913-0.963)* | 0.863 | 0.875 |
| | | LUV-Net(Resnet-50)† | 0.870 (0.834-0.907)* | 0.766 (0.716-0.816)* | 0.744 (0.691-0.797)* | 0.742 (0.655-0.830)* | 0.768 | 0.783 |
| | | LUV-Net(Densenet-161)† | **0.924 (0.895-0.953)** | **0.824 (0.782-0.867)** | 0.837 (0.795-0.880) | 0.967 (0.949-0.985) | **0.886** | **0.890** |
| | | LUV-Net(Densenet-161) | 0.897 (0.863-0.931) | 0.601 (0.544-0.659)* | **0.898 (0.869-0.927)*** | 0.726 (0.651-0.801)* | 0.805 | 0.783 |

Table 5: Development set results, * indicates that the P-value is less than 0.05, and †
indicates ImageNet pretrained model.

## A.2. Results on the temporally separated validation set

| Fold # | Input Type | Model | AUC | | | | Avg | |
|---|---|---|---|---|---|---|---|---|
| | | | A-line | B-line | Consolidation | Pleural effusion | Micro | Macro |
| Fold 0 | Frame (Image) | frame-based | 0.763 (0.683-0.788)* | **0.899 (0.867-0.930)*** | 0.773 (0.717-0.829) | 0.793 (0.656-0.931)* | 0.807 | 0.801 |
| | Video | MViT-B | 0.592 (0.523-0.662)* | 0.612 (0.557-0.668)* | 0.606 (0.541-0.671)* | 0.699 (0.594-0.804)* | 0.690 | 0.629 |
| | | Swin-B | 0.674 (0.611-0.736) | 0.579 (0.521-0.636) | 0.698 (0.634-0.762) | 0.737 (0.639-0.836) | 0.715 | 0.674 |
| | | C3D | 0.796 (0.739-0.852) | 0.735 (0.683-0.787)* | 0.745 (0.670-0.820) | 0.868 (0.793-0.942) | 0.865 | 0.788 |
| | | R2Plus1D | 0.680 (0.604-0.756)* | 0.626 (0.571-0.682)* | 0.716 (0.647-0.786)* | 0.854 (0.761-0.947) | 0.706 | 0.721 |
| | | CNN(Densenet-161)†+LSTM | 0.374 (0.295-0.452)* | 0.470 (0.413-0.527)* | 0.795 (0.742-0.848) | 0.781 (0.666-0.895)* | 0.550 | 0.606 |
| | | USVN(Densenet-161)† | 0.803 (0.759-0.847) | 0.874 (0.838-0.910)* | **0.842 (0.801-0.883)*** | 0.775 (0.633-0.917)* | 0.837 | 0.825 |
| | | LUV-Net(Resnet-50)† | 0.741 (0.691-0.792)* | 0.776 (0.730-0.821)* | 0.808 (0.759-0.857) | 0.832 (0.750-0.913)* | 0.810 | 0.791 |
| | | LUV-Net(Densenet-161)† | 0.819 (0.775-0.863) | 0.847 (0.809-0.885) | 0.778 (0.723-0.833) | **0.877 (0.803-0.930)*** | **0.882** | **0.832** |
| | | LUV-Net(Densenet-161) | **0.848 (0.803-0.893)** | 0.760 (0.713-0.808)* | 0.777 (0.713-0.840) | 0.863 (0.780-0.946) | 0.793 | 0.814 |
| Fold 1 | Frame (Image) | frame-based | 0.790 (0.744-0.836)* | **0.889 (0.857-0.921)*** | 0.751 (0.687-0.815)* | 0.851 (0.752-0.950) | 0.824 | 0.821 |
| | Video | MViT-B | 0.747 (0.686-0.810)* | 0.560 (0.504-0.617)* | 0.697 (0.632-0.762)* | 0.930 (0.882-0.978) | 0.844 | 0.735 |
| | | Swin-B | 0.715 (0.656-0.775)* | 0.582 (0.525-0.638)* | 0.735 (0.668-0.802)* | 0.925 (0.879-0.971) | 0.827 | 0.741 |
| | | C3D | 0.785 (0.726-0.844)* | 0.688 (0.636-0.740)* | 0.749 (0.677-0.821) | 0.877 (0.802-0.953) | **0.853** | 0.777 |
| | | R2Plus1D | 0.575 (0.496-0.655)* | 0.638 (0.584-0.693)* | 0.664 (0.591-0.736)* | 0.875 (0.802-0.947) | 0.710 | 0.690 |
| | | CNN(Densenet-161)†+LSTM | 0.328 (0.255-0.401)* | 0.464 (0.407-0.521)* | 0.759 (0.699-0.820) | 0.832 (0.734-0.929)* | 0.577 | 0.597 |
| | | USVN(Densenet-161)† | 0.774 (0.686-0.803)* | 0.772 (0.726-0.818)* | **0.844 (0.803-0.885)** | 0.838 (0.749-0.926)* | 0.760 | 0.801 |
| | | LUV-Net(Resnet-50)† | 0.767 (0.713-0.821)* | 0.744 (0.670-0.792)* | 0.828 (0.779-0.878) | **0.911 (0.865-0.957)** | 0.845 | 0.814 |
| | | LUV-Net(Densenet-161)† | **0.893 (0.856-0.929)** | **0.835 (0.794-0.875)** | 0.802 (0.746-0.859) | 0.907 (0.839-0.975) | 0.825 | **0.861** |
| | | LUV-Net(Densenet-161) | 0.845 (0.796-0.893)* | 0.782 (0.737-0.827)* | 0.724 (0.651-0.797)* | 0.884 (0.809-0.959) | 0.768 | 0.810 |
| Fold 2 | Frame (Image) | frame-based | 0.775 (0.728-0.822)* | **0.898 (0.867-0.930)*** | 0.796 (0.740-0.853) | 0.805 (0.682-0.928)* | 0.837 | **0.838** |
| | Video | MViT-B | 0.478 (0.400-0.556)* | 0.510 (0.453-0.567)* | 0.572 (0.505-0.640)* | 0.733 (0.658-0.809)* | 0.755 | 0.576 |
| | | Swin-B | 0.704 (0.647-0.761) | 0.548 (0.490-0.606) | 0.550 (0.479-0.620) | 0.654 (0.550-0.759) | 0.697 | 0.616 |
| | | C3D | 0.761 (0.695-0.828) | 0.617 (0.562-0.672)* | 0.733 (0.662-0.803) | 0.835 (0.741-0.930)* | 0.809 | 0.739 |
| | | R2Plus1D | 0.718 (0.642-0.795) | 0.638 (0.581-0.695)* | 0.656 (0.583-0.730)* | 0.875 (0.807-0.943) | 0.723 | 0.724 |
| | | CNN(Densenet-161)†+LSTM | 0.229 (0.170-0.288)* | 0.562 (0.505-0.619)* | 0.794 (0.742-0.846) | 0.822 (0.722-0.923)* | 0.771 | 0.603 |
| | | USVN(Densenet-161)† | 0.783 (0.739-0.826) | 0.835 (0.794-0.876) | 0.806 (0.752-0.861) | 0.874 (0.788-0.959) | 0.802 | 0.826 |
| | | LUV-Net(Resnet-50)† | 0.799 (0.751-0.847) | 0.760 (0.713-0.807)* | **0.829 (0.781-0.877)*** | 0.909 (0.859-0.958) | 0.828 | 0.826 |
| | | LUV-Net(Densenet-161)† | 0.771 (0.725-0.818) | 0.862 (0.825-0.899) | 0.781 (0.728-0.834) | 0.884 (0.807-0.961) | **0.848** | 0.826 |
| | | LUV-Net(Densenet-161) | **0.800 (0.749-0.850)** | 0.835 (0.794-0.875) | 0.727 (0.654-0.799)* | **0.930 (0.886-0.974)*** | 0.727 | 0.824 |
| Fold 3 | Frame (Image) | frame-based | 0.748 (0.719-0.818)* | 0.890 (0.865-0.927)* | 0.763 (0.759-0.868) | 0.799 (0.628-0.921)* | 0.763 | 0.801 |
| | Video | MViT-B | 0.790 (0.734-0.846)* | 0.516 (0.458-0.573)* | 0.679 (0.607-0.751)* | 0.945 (0.907-0.982)* | 0.822 | 0.734 |
| | | Swin-B | 0.681 (0.619-0.744)* | 0.557 (0.499-0.616)* | 0.657 (0.589-0.725)* | 0.765 (0.680-0.850)* | 0.711 | 0.667 |
| | | C3D | 0.805 (0.746-0.864)* | 0.741 (0.688-0.795)* | 0.762 (0.692-0.833)* | **0.905 (0.846-0.964)*** | 0.841 | 0.805 |
| | | R2Plus1D | 0.647 (0.566-0.728)* | 0.645 (0.591-0.700)* | 0.711 (0.644-0.777)* | 0.858 (0.722-0.944) | 0.786 | 0.717 |
| | | CNN(Densenet-161)†+LSTM | 0.444 (0.364-0.523)* | 0.519 (0.462-0.577) | 0.737 (0.677-0.797)* | 0.754 (0.648-0.859)* | 0.543 | 0.615 |
| | | USVN(Densenet-161)† | 0.804 (0.759-0.849)* | 0.865 (0.828-0.901) | **0.874 (0.834-0.915)*** | 0.806 (0.684-0.927) | 0.849 | 0.839 |
| | | LUV-Net(Resnet-50)† | 0.809 (0.762-0.856)* | 0.768 (0.721-0.815)* | 0.825 (0.779-0.871) | 0.895 (0.833-0.956)* | 0.835 | 0.826 |
| | | LUV-Net(Densenet-161)† | **0.895 (0.860-0.930)** | **0.878 (0.844-0.912)** | 0.825 (0.733-0.877) | 0.829 (0.727-0.932) | **0.874** | **0.858** |
| | | LUV-Net(Densenet-161) | 0.820 (0.766-0.874)* | 0.727 (0.676-0.778)* | 0.748 (0.679-0.817)* | 0.822 (0.712-0.933) | 0.819 | 0.781 |
| Fold 4 | Frame (Image) | frame-based | 0.739 (0.687-0.790)* | 0.888 (0.845-0.922) | 0.775 (0.711-0.838) | 0.814 (0.691-0.938)* | 0.813 | 0.805 |
| | Video | MViT-B | 0.558 (0.482-0.634)* | 0.492 (0.434-0.550)* | 0.610 (0.546-0.673)* | 0.820 (0.753-0.887)* | 0.588 | 0.622 |
| | | Swin-B | 0.437 (0.363-0.511)* | 0.462 (0.405-0.520)* | 0.520 (0.448-0.592)* | 0.675 (0.582-0.768)* | 0.734 | 0.526 |
| | | C3D | 0.828 (0.778-0.878) | 0.732 (0.679-0.784)* | 0.745 (0.672-0.818) | 0.927 (0.880-0.974)* | 0.870 | 0.810 |
| | | R2Plus1D | 0.673 (0.584-0.751)* | 0.621 (0.565-0.677)* | 0.674 (0.601-0.747)* | 0.880 (0.805-0.956) | 0.745 | 0.714 |
| | | CNN(Densenet-161)†+LSTM | 0.453 (0.380-0.527)* | 0.442 (0.385-0.500)* | 0.693 (0.633-0.752)* | 0.580 (0.456-0.704)* | 0.571 | 0.544 |
| | | USVN(Densenet-161)† | 0.809 (0.766-0.853) | **0.893 (0.860-0.926)** | **0.865 (0.824-0.905)*** | **0.930 (0.880-0.979)*** | **0.874** | **0.875** |
| | | LUV-Net(Resnet-50)† | 0.833 (0.788-0.878) | 0.773 (0.727-0.820)* | 0.831 (0.783-0.878) | 0.838 (0.748-0.928) | 0.861 | 0.820 |
| | | LUV-Net(Densenet-161)† | 0.798 (0.752-0.843) | 0.890 (0.857-0.923) | 0.811 (0.755-0.868) | 0.870 (0.786-0.954) | 0.859 | 0.844 |
| | | LUV-Net(Densenet-161) | **0.890 (0.848-0.931)*** | 0.786 (0.741-0.830)* | 0.742 (0.671-0.813)* | 0.927 (0.882-0.973)* | 0.870 | 0.838 |

Table 6: Temporally separated set results. * indicates that the P-value is less than 0.05, and † indicates ImageNet pretrained model.

### A.3. Results of temporal feature extraction study on development dataset

To validate the effectiveness of our temporal feature extraction module, we conducted an ablation study by comparing LUVM with its variant without temporal feature extraction (LUVM w/o temporal) across all five folds on the development dataset. Table 7 presents the detailed results of this comparison. The results demonstrate that the temporal feature extraction module generally contributes to improved performance, though the magnitude of improvement varies across different patterns. For A-line detection, both variants showed comparable performance, with LUVM showing slight improvements in Folds 0-2 (e.g., Fold 2: 0.933 vs 0.910) but marginally lower performance in Folds 3-4. This suggests that A-line patterns may be less dependent on temporal information for accurate detection. More notable improvements were observed in B-line detection, particularly in Folds 0 and 2, where LUVM achieved AUC scores of 0.841 and 0.827 compared to 0.827 and 0.797 for the non-temporal variant, respectively. The temporal feature extraction seemed particularly beneficial for Consolidation pattern detection, with consistent improvements across most folds and statistical significance observed in Fold 3 (0.882 vs 0.857, p¡0.05). Pleural Effusion detection showed interesting results, with the temporal feature extraction module contributing to statistically significant improvements in several folds (Folds 0 and 3). This suggests that temporal information plays a crucial role in accurately identifying this particular pattern. In terms of overall performance metrics, LUVM consistently achieved higher or comparable micro and macro averages across all folds compared to its non-temporal variant. The most substantial improvements were observed in Fold 2, where both Micro (0.890 vs 0.867) and Macro (0.890 vs 0.873) averages showed clear advantages of temporal feature extraction. These results validate the effectiveness of our temporal feature extraction module in capturing dynamic pattern characteristics while maintaining robust performance across different data splits.

| Fold # | Model | AUC | | | | Avg | |
|---|---|---|---|---|---|---|---|
| | | A-line | B-line | Consolidation | Pleural effusion | Micro | Macro |
| Fold 0 | LUV-Net (w/o temporal) | 0.894 (0.855-0.934) | 0.827 (0.786-0.869) | 0.832 (0.793-0.871) | 0.955 (0.935-0.975)* | 0.867 | 0.879 |
| | LUV-Net (w/ temporal) | **0.900 (0.860-0.940)** | **0.841 (0.802-0.880)** | **0.851 (0.810-0.892)** | **0.974 (0.959-0.989)** | **0.880** | **0.893** |
| Fold 1 | LUV-Net (w/o temporal) | 0.891 (0.849-0.933)* | 0.819 (0.776-0.863) | **0.863 (0.827-0.900)** | 0.944 (0.923-0.966) | **0.883** | 0.881 |
| | LUV-Net (w/ temporal) | **0.914 (0.886-0.943)** | **0.821 (0.779-0.862)** | 0.860 (0.822-0.897) | **0.950 (0.927-0.972)** | 0.882 | **0.887** |
| Fold 2 | LUV-Net (w/o temporal) | 0.910 (0.875-0.944)* | 0.797 (0.753-0.842)* | 0.817 (0.774-0.860) | 0.961 (0.940-0.981) | 0.867 | 0.873 |
| | LUV-Net (w/ temporal) | **0.933 (0.908-0.958)** | **0.827 (0.786-0.868)** | **0.834 (0.796-0.871)** | **0.963 (0.944-0.983)** | **0.890** | **0.890** |
| Fold 3 | LUV-Net (w/o temporal) | **0.927 (0.898-0.955)** | 0.845 (0.805-0.885) | 0.857 (0.816-0.897)* | 0.968 (0.952-0.985)* | 0.901 | 0.901 |
| | LUV-Net (w/ temporal) | 0.919 (0.891-0.948) | **0.855 (0.817-0.893)** | **0.882 (0.848-0.916)** | **0.975 (0.958-0.992)** | **0.903** | **0.909** |
| Fold 4 | LUV-Net (w/o temporal) | **0.927 (0.896-0.957)** | **0.840 (0.798-0.883)** | **0.838 (0.797-0.879)** | 0.952 (0.932-0.971)* | 0.884 | **0.891** |
| | LUV-Net (w/ temporal) | 0.924 (0.895-0.953) | 0.824 (0.782-0.867) | 0.837 (0.795-0.880) | **0.967 (0.949-0.985)** | **0.886** | 0.890 |

Table 7: 5-fold cross validation results of temporal feature extraction study, * indicates that the P-value is less than 0.05

## Appendix B. Qualitative Analysis

To further investigate the interpretability of the proposed LUV-Net model, we performed qualitative analysis by visualizing the attention scores across video frames for multiple labels and highlighting the most informative frames. The attention mechanism incorporated in our model provides a pathway to understand which frames contribute the most to the classification of each pattern. This section presents the results of this analysis, supported by both visual plots and mathematical expressions.

The proposed model applies a pattern-specific attention mechanism, where each LUS pattern $p \in \{p_0, p_1, p_2, p_3\}$ corresponds to A-line, B-line, consolidation, and pleural effusion. For each pattern $p$, a set of query vectors $\{q_k^p\}_{k=1}^{N_a}$ is learned independently for its corresponding attention heads. The attention score for frame $t$ and head $k$ is computed as the dot product between the frame-level representation $h_{t,k}$ and the query vector $q_k^p$:

$$\lambda_{t,k}^p = h_{t,k} \cdot q_k^p \tag{5}$$

These scores are scaled by $\sqrt{d_k}$ and normalized across the temporal dimension using the softmax function to produce attention weights:

$$a_{t,k}^p = \frac{\exp\left(\lambda_{t,k}^p / \sqrt{d_k}\right)}{\sum_{t'=1}^{L} \exp\left(\lambda_{t',k}^p / \sqrt{d_k}\right)} \tag{6}$$

To visualize the attention distribution across frames for each pattern, we aggregate the attention weights from all $N_a$ heads:

$$\hat{\alpha}_t^p = \sum_{k=1}^{N_a} a_{t,k}^p \tag{7}$$

To ensure comparability of attention scores across different videos, we apply Min-Max normalization to the aggregated attention scores:

$$\alpha_t^{p,\text{norm}} = \frac{\hat{\alpha}_t^p - \min_{t'}(\hat{\alpha}_{t'}^p)}{\max_{t'}(\hat{\alpha}_{t'}^p) - \min_{t'}(\hat{\alpha}_{t'}^p)} \tag{8}$$

$\alpha_t^{p,\text{norm}} \in [0, 1]$ represents the normalized importance of frame $t$ for pattern $p$. This normalized score allows for frame-level visualization of model attention, where the most relevant frames for each pattern are highlighted based on their contribution to the corresponding prediction.

## Appendix C. Ablation study

To evaluate the effectiveness of our proposed LUV-Net model and understand the impact of various architectural choices, we conducted extensive ablation studies. We aimed to identify the optimal parameters for both the temporal feature extraction network and the spatial feature extraction network. Specifically, we experimented with the kernel size of the temporal feature extraction network and the number of attention heads in the spatial feature extraction network.

### C.1. Effect of Kernel size of temporal feature extraction network

The analysis of different 1D kernel size (ranging from 1 to 29) on the development set revealed interesting patterns in model performance (Table 8). The results reveal that a kernel size of 13 achieves optimal overall performance. Specifically, with kernel size 13, we observe strong performance across all evaluation metrics: 0.900 for A-line detection, 0.853 for B-line detection, 0.882 for Consolidation detection, and 0.975 for Pleural effusion

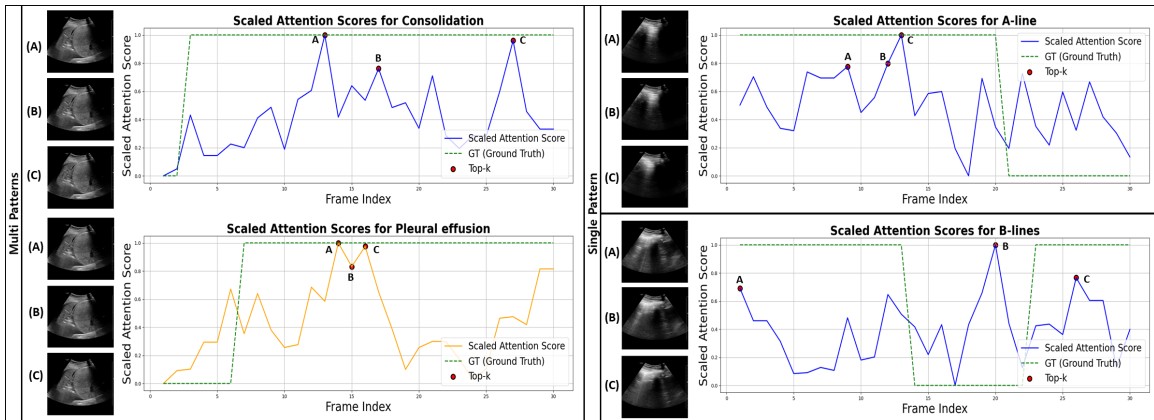

Figure 4: Visualization of attention scores and corresponding top-3 frames for each patterns on temporally separated set.

detection. The macro and micro averages at kernel size 13 are 0.908 and 0.902 respectively, indicating balanced performance across all classes. Based on these findings, we selected a kernel size of 13 as the most effective configuration for the temporal feature extraction network.

| | | Kernel size | | | | | | | | | | | | | |
|---|---|---|---|---|---|---|---|---|---|---|---|---|---|---|---|
| | | 1 | 3 | 5 | 7 | 9 | 11 | 13 | 15 | 17 | 19 | 21 | 23 | 25 | 27 | 29 |
| AUC | A-line | 0.918 | 0.928 | 0.912 | 0.909 | 0.906 | 0.909 | 0.917 | 0.900 | 0.912 | 0.909 | 0.901 | 0.926 | 0.918 | 0.922 | 0.922 |
| | B-line | 0.827 | 0.857 | 0.855 | 0.847 | 0.826 | 0.785 | 0.853 | 0.823 | 0.811 | 0.806 | 0.816 | 0.820 | 0.801 | 0.818 | 0.816 |
| | Consolidation | 0.862 | 0.856 | 0.861 | 0.846 | 0.876 | 0.870 | 0.882 | 0.880 | 0.884 | 0.884 | 0.860 | 0.897 | 0.892 | 0.885 | 0.881 |
| | Pleural effusion | 0.965 | 0.969 | 0.965 | 0.963 | 0.977 | 0.971 | 0.975 | 0.975 | 0.972 | 0.968 | 0.973 | 0.972 | 0.975 | 0.965 | 0.964 |
| Avg | Micro | 0.891 | 0.902 | 0.886 | 0.888 | 0.894 | 0.889 | 0.902 | 0.893 | 0.890 | 0.890 | 0.874 | 0.907 | 0.902 | 0.888 | 0.888 |
| | Macro | 0.894 | 0.904 | 0.899 | 0.893 | 0.898 | 0.885 | 0.908 | 0.896 | 0.896 | 0.893 | 0.889 | 0.905 | 0.898 | 0.899 | 0.897 |

Table 8: Development set Kernel size

## C.2. Effect of number of attention heads of spatial feature extraction network

To investigate the optimal number of attention heads in our spatial feature extraction network, we conducted experiments varying the number of attention heads from 1 to 96. Table 9 presents the performance evaluation across different metrics for each attention head configuration. Our experimental results indicate that the number of attention heads has a relatively stable impact on model performance. When examining the results, we observe that using 8 attention heads achieves optimal performance across most metrics. With 8 attention heads, the model demonstrates strong results with AUC scores of 0.917 for A-line detection, 0.855 for B-line detection, 0.882 for Consolidation detection, and 0.975 for Pleural effusion detection. Furthermore, both micro and macro averages achieved the best overall performance (Micro: 0.903, Macro: 0.909) with 8 attention heads. Interestingly, increasing the number of attention heads beyond 8 does not yield significant performance improvements. Similarly, using fewer attention heads (1, 2, 4) shows marginally lower performance across most metrics.

| | | Attn head num | | | | | | |
|---|---|---|---|---|---|---|---|---|
| | | 1 | 2 | 4 | 8 | 16 | 32 | 96 |
| AUC | A-line | 0.914 | 0.919 | 0.918 | 0.919 | 0.917 | 0.917 | 0.915 |
| | B-line | 0.852 | 0.854 | 0.851 | 0.855 | 0.852 | 0.853 | 0.850 |
| | Consolidation | 0.881 | 0.882 | 0.880 | 0.882 | 0.880 | 0.882 | 0.881 |
| | Pleural effusion | 0.974 | 0.975 | 0.973 | 0.975 | 0.975 | 0.975 | 0.976 |
| Avg | Micro | 0.901 | 0.902 | 0.897 | 0.903 | 0.901 | 0.902 | 0.901 |
| | Macro | 0.907 | 0.909 | 0.907 | 0.909 | 0.907 | 0.908 | 0.907 |

Table 9: Development set Number of Attn head

