# OpenReview forum: "LUV-Net: Multi-Pattern Lung Ultrasound Video Classification through Pattern-Specific Attention with Efficient Temporal Feature Extraction"
_MIDL.io/2025/Conference — MIDL 2025 Poster_

### Official Review · Reviewer_Q4am · 2025-02-15

**Confidence:** 5
**Preliminary Rating:** 3
**Final Rating:** 4

**Summary:**

The authors propose to modify USVN with separate attention mechanisms for different LUS patterns instead of applying uniform attention mechanisms. The authors conduct evaluations of their method with internal datasets to demonstrate improvement compared to previous studies.

**Strengths:**

(1) The authors conduct cross-validation and demonstrate the results in detail. The authors also share test results for both the development set and the temporally separated validation set.

(2) The authors also share their code to ensure the reproducibility of their work.

**Weaknesses:**

(1) Lack of comparison with other state-of-the-art video action recognition methods. Methods such as C3D, R2Plus1D, and CNN+LSTM were developed for video action recognition at least seven years ago. Please compare with more recent state-of-the-art video action recognition/video classification methods instead, especially transformer-based methods.

(2) USVN used ResNet50 as the backbone, while the proposed method used DenseNet-161. Please make sure the same backbone is used for a fair comparison between these two methods.

(3) R2Plus1D outperforms C3D on video action recognition/video classification. However, the results reported by the authors state otherwise. Please ensure the training of R2Plus1D is set up correctly.

**Detailed Comments:**

For weakness point (1), please refer to Kinetics datasets to select state-of-the-art video action recognition/video classification methods.

For weakness point (2), the authors mentioned in the paper: "For our model architecture, we employed the ImageNet pre-trained DenseNet-161 as the encoder, which was also used in the CNN+LSTM and frame-based method." Please ensure DenseNet-161 is used for USVN.

For weakness point (3), please check the following papers to ensure the training of R2Plus1D is set up correctly.

[1] Tran D, Wang H, Torresani L, et al. A closer look at spatiotemporal convolutions for action recognition[C]//Proceedings of the IEEE conference on Computer Vision and Pattern Recognition. 2018: 6450-6459.

[2] Ghadiyaram D, Tran D, Mahajan D. Large-scale weakly-supervised pre-training for video action recognition[C]//Proceedings of the IEEE/CVF conference on computer vision and pattern recognition. 2019: 12046-12055.

Please conduct a typo check for the paper, for example:

(1) "Data collection and Annotation" to "Data Collection and Annotation"

**Justification Of The Final Rating:**

I sincerely thank the authors for their thoughtful response. The inclusion of additional experiments and model performance analysis have significantly elevated the quality of the paper. As a result, I am happy to adjust my rating from borderline to weak accept.

**Justification Of The Preliminary Rating:**

As the paper lacks comparisons with state-of-the-art methods and other potential evaluation issues, I vote for Boardline. I encourage the authors to revise the paper, especially for the part of comparison with state-of-the-art methods.

**Questions To Address In The Rebuttal:**

Please address weak points (1) and (2).

For weakness point (3), if the training is set up correctly with adequate pre-training and the results for R2Plus1D are still bad, please discuss the potential reasons why C3D outperforms R2Plus1D in the paper.

**Special Issue:**

No

---

> ### Author Response · Authors · 2025-03-07
> **Official Reply to Reviewer Q4am**
>
> We sincerely appreciate the detailed and constructive feedback provided by the reviewer. Your comments have significantly helped us improve our work.
>
> **1) Comparison with State-of-the-Art Video Models**
>
> We appreciate the reviewer's concern regarding comparisons with more recent state-of-the-art models. Our baseline selection was driven by the prevalent use of conventional architectures (R2Plus1D, C3D, CNN+LSTM) in LUS research, while direct comparison with LUS-specific models was challenging due to lack of publicly available code.
>
> As suggested by the reviewer, we conducted additional experiments with ViT-based MViT and Video Swin Transformer from Torchvision. However, since differences in frame resolution and clip length prevented the use of pretrained weights, we trained these models from scratch under identical conditions. The results are included in Tables 1 and 2 of the revised manuscript, with comparative analysis between baseline models and our proposed approach detailed in the Discussion section.
>
> **2) USVN Backbone Consistency**
>
> We sincerely apologize for any ambiguity in the initial manuscript regarding the backbone selection for USVN. We acknowledge that the description may have led to confusion, and we appreciate the reviewer for bringing this to our attention.
>
> To clarify, all experiments involving USVN were conducted using the same backbone, DenseNet-161, to ensure a fair and consistent comparison across models. This ensures that differences in performance are attributable to the model architecture itself, rather than variations in feature extraction quality. We have revised the manuscript to explicitly state this, ensuring that readers clearly understand the experimental setup.
>
> We are grateful to the reviewer for highlighting this issue, as it has helped us improve the clarity and precision of our paper.
>
> **3) Training and Performance of R2Plus1D**
>
> We appreciate your concern regarding the reported performance difference between C3D and R2Plus1D in our LUS classification task.
> Following your comment, we carefully reviewed our R2Plus1D implementation code and training process to confirm there were no issues with our methodology.
> While R2Plus1D typically outperforms C3D in general video action recognition tasks, we observed that C3D achieves superior results in our specific LUS classification setting under identical training conditions and standard protocols.
>
>    * **Differences Between Natural Video Datasets and LUS Data**
>
> In conventional natural video datasets, objects and specific motions tend to be distributed throughout the entire video sequence, leading to strong temporal dependencies across all frames. This characteristic makes models like R2Plus1D, which are optimized for capturing long-term motion patterns, highly effective in action recognition tasks.
> However, LUS videos exhibit fundamentally different temporal dynamics. Key diagnostic patterns, such as A-lines, B-lines, and consolidations, do not persist across all frames but instead appear in specific segments of the sequence.
> Since diagnostic patterns in LUS are localized to specific frames, not all frames contain equally informative content for classification. Given this distinction, a model that emphasizes local frame-to-frame dependencies rather than long-range temporal patterns may be better suited for LUS analysis.
>
>   * **Comparison Between C3D and R2Plus1D**
>
>     * C3D applies 3D convolutions to jointly learn spatial and temporal features, effectively capturing cross-frame spatial dependencies. In contrast, R2Plus1D separates this process into 2D spatial convolutions followed by 1D temporal convolutions, optimizing it for long-term motion modeling.
>     * C3D is more effective than R2Plus1D for LUS classification tasks, as it captures key patterns that appear within a short time window, where diagnostic features emerge over a few consecutive frames. In contrast, R2Plus1D, which is designed for long-term motion modeling, may be less optimal for LUS due to its more localized temporal dependencies.
>
> These findings suggest that C3D’s unified spatiotemporal learning is particularly advantageous for LUS video classification, which explains why it demonstrated better performance than the R2Plus1D model in our study.

---

> > ### Comment · Reviewer_Q4am · 2025-03-13
> > **Please conduct fair comparisions with different methods.**
> >
> > I would like to extend my gratitude to the authors for their responses. However, it is important to note that comparing models that utilize pre-trained weights with those trained from scratch is not a fair assessment. For a fair comparison, C3D, R2PLUS1D, MVIT-B, and Swin-B, which are all trained from scratch, should be compared with LUV-Net, which is trained from scratch without using pre-trained weights. Please revise your result tables to reflect these fair comparisons.

---

> > ### Author Response · Authors · 2025-03-13
> > **Official Reply to Reviewer Q4am**
> >
> > We sincerely appreciate your continued effort in reviewing our work and providing insightful comments.
> >
> > To maintain consistency with the previous USVN study [1], which employed ImageNet pre-trained models, we also used ImageNet pre-trained weights for all 2D CNN-based models, including LUV-Net, CNN+LSTM, and USVN. This approach was chosen to ensure comparability between our proposed model and the prior work, thereby aligning with the established experimental framework. Additionally, while using pre-trained weights may not substantially improve performance in medical imaging tasks, it has been shown to accelerate convergence due to better weight scaling, leading to faster training [2]. Based on these reasons, we did not anticipate that using pre-trained models would have a critical impact on our results.
> >
> > However, to ensure a fully fair comparison without using pre-trained weights, we would need to train and evaluate all 2D CNN-based models (including LUV-Net, CNN+LSTM, and USVN) from scratch under a 5-fold cross-validation setting. This process would require significantly more time for convergence due to the increased training complexity. Given the Discussion Period deadline (8 March – 14 March 2025), we will update the manuscript with as many results as possible before this stage and share them during the discussion. For additional experiments that cannot be completed within this timeframe, we will make our best effort to include them in the final camera-ready version of the paper.
> >
> > [1] Smith, D. H., Lineberger, J. P., & Baker, G. H. (2023, October). On the relevance of temporal features for medical ultrasound video recognition. In International Conference on Medical Image Computing and Computer-Assisted Intervention (pp. 744-753). Cham: Springer Nature Switzerland.
> >
> > [2] Raghu, M., Zhang, C., Kleinberg, J., & Bengio, S. (2019). Transfusion: Understanding transfer learning for medical imaging. Advances in Neural Information Processing Systems, 32.

---

> > > ### Comment · Reviewer_Q4am · 2025-03-13
> > > **Follow up: Please conduct fair comparisions with different methods**
> > >
> > > Thank you for your response. However, the notion that transfer learning may not significantly enhance performance in your task is unpersuasive.
> > >
> > > In the Transfusion paper, the authors utilized two extensive datasets: the Retina dataset with 128,175 retinal images and the CheXpert dataset with 224,316 chest radiographs. Your dataset, in contrast, contains only around 3,000 samples. Given that you are training temporal models, the total frame count does not multiply. Consider the substantial amount of data required to train models like C3D, R(2+1)D, MVIT-B, and Swin-B. Pre-training is crucial for these models. For instance, the K400 dataset comprises over 300,000 video clips. Pre-training on a large scale can also significantly boost the performance of various action recognition models [1].
> > >
> > > I did not suggest conducting all 2D CNN-based models from scratch. I only request that you perform one more experiment with LUV-Net, trained from scratch. Please add the results to the table and modify the table to clearly differentiate between models trained from scratch and those that used transfer learning for a fair comparison.
> > >
> > > [1] Ghadiyaram D, Tran D, Mahajan D. Large-scale weakly-supervised pre-training for video action recognition[C]//Proceedings of the IEEE/CVF conference on computer vision and pattern recognition. 2019: 12046-12055.

---

> > ### Author Response · Authors · 2025-03-14
> > **Official Reply to Reviewer Q4am**
> >
> > We appreciate the reviewer's insightful comments regarding the comparison between pretrained and non-pretrained models. To address this concern, we have included additional results evaluating LUV-Net without using pretrained weights, ensuring a fair comparison with non-pretrained baselines (C3D, R2Plus1D, MViT-B, and Swin-B). While the performance of LUV-Net without pretraining is lower than its pretrained counterpart, the results indicate no critical degradation in classification ability. Furthermore, even without pretraining, LUV-Net consistently outperforms the non-pretrained baseline models, particularly excelling in A-line and pleural effusion detection.
> >
> > **Note:** † indicates models that **used pretrained weights**.
> > ## **Results on the Development Set**
> > | Input Type  | Model         | Backbone                        | A-line       | B-line       | Consolidation | Pleural Effusion | Micro       | Macro       |
> > |------------|--------------|---------------------------------|--------------|--------------|--------------|-----------------|--------------|--------------|
> > | Frame (Image) | Frame-based  | CNN-based (DenseNet-161)†      | **0.926±0.007** | 0.800±0.033  | 0.831±0.018  | **0.973±0.004** | 0.881±0.008  | 0.883±0.011  |
> > | Video      | MViT-B       | Transformer-based              | 0.598±0.085  | 0.527±0.130  | 0.620±0.066  | 0.752±0.109  | 0.619±0.055  | 0.626±0.066  |
> > |            | Swin-B       | Transformer-based              | 0.582±0.095  | 0.574±0.130  | 0.604±0.077  | 0.714±0.100  | 0.601±0.064  | 0.620±0.074  |
> > |            | C3D         | CNN-based                      | 0.789±0.062  | 0.626±0.038  | 0.788±0.023  | 0.930±0.002  | 0.791±0.025  | 0.792±0.023  |
> > |            | R2Plus1D    | CNN-based                      | 0.726±0.051  | 0.619±0.078  | 0.801±0.031  | 0.830±0.071  | 0.720±0.031  | 0.746±0.035  |
> > |            | CNN+LSTM    | CNN-based (DenseNet-161)†      | 0.420±0.066  | 0.353±0.032  | 0.737±0.043  | 0.911±0.042  | 0.587±0.023  | 0.607±0.007  |
> > |            | USVN        | CNN-based (DenseNet-161)†      | 0.879±0.080  | 0.770±0.137  | **0.879±0.080**  | 0.931±0.037  | 0.846±0.062  | 0.856±0.058  |
> > |            | LUV-Net     | CNN-based (ResNet-50)†         | 0.867±0.026  | 0.778±0.020  | 0.762±0.025  | 0.806±0.140  | 0.808±0.030  | 0.806±0.034  |
> > |            | LUV-Net     | CNN-based (DenseNet-161)†      | 0.918±0.013  | **0.834±0.014**  | 0.853±0.019  | 0.966±0.010  | **0.888±0.009**  | **0.894±0.009**  |
> > |            | LUV-Net     | CNN-based (DenseNet-161)       | 0.865±0.026  | 0.644±0.059  | 0.872±0.033  | 0.849±0.098  | 0.809±0.045  | 0.809±0.034  |
> >
> > ## **Results on the Temporally Separated Set**
> > | Input Type  | Model         | Backbone                        | A-line       | B-line       | Consolidation | Pleural Effusion | Micro       | Macro       |
> > |------------|--------------|---------------------------------|--------------|--------------|--------------|-----------------|--------------|--------------|
> > | Frame (Image) | Frame-based  | CNN-based (DenseNet-161)†      | 0.763±0.025  | **0.893±0.005** | 0.772±0.019  | 0.812±0.023  | 0.809±0.031  | 0.813±0.014  |
> > | Video      | MViT-B       | Transformer-based              | 0.633±0.118  | 0.538±0.043  | 0.633±0.047  | 0.825±0.100  | 0.740±0.093  | 0.659±0.064  |
> > |            | Swin-B       | Transformer-based              | 0.642±0.104  | 0.546±0.044  | 0.632±0.083  | 0.751±0.096  | 0.737±0.047  | 0.645±0.071  |
> > |            | C3D         | CNN-based                      | 0.795±0.030  | 0.703±0.047  | 0.747±0.010  | 0.882±0.032  | 0.848±0.025  | 0.784±0.025  |
> > |            | R2Plus1D    | CNN-based                      | 0.659±0.056  | 0.634±0.010  | 0.684±0.027  | 0.868±0.011  | 0.734±0.030  | 0.713±0.014  |
> > |            | CNN+LSTM    | CNN-based (DenseNet-161)†      | 0.366±0.092  | 0.491±0.047  | 0.756±0.042  | 0.754±0.084  | 0.602±0.069  | 0.593±0.029  |
> > |            | USVN        | CNN-based (DenseNet-161)†      | 0.795±0.024  | 0.848±0.039  | **0.846±0.022**  | 0.845±0.068  | 0.824±0.049  | 0.833±0.027  |
> > |            | LUV-Net     | CNN-based (ResNet-50)†         | 0.790±0.032  | 0.764±0.011  | 0.824±0.008  | 0.877±0.035  | 0.836±0.017  | 0.815±0.013  |
> > |            | LUV-Net     | CNN-based (DenseNet-161)†      | 0.835±0.057  | 0.862±0.022  | 0.799±0.021  | 0.873±0.026  | **0.858±0.023**  | **0.844±0.015**  |
> > |            | LUV-Net     | CNN-based (DenseNet-161)       | **0.841±0.030**  | 0.778±0.035  | 0.744±0.019  | **0.885±0.041**  | 0.795±0.048  | 0.813±0.019  |
> >
> > These findings further support the robustness and effectiveness of our proposed model architecture. The revised results have been integrated into the manuscript accordingly.

---

> > > ### Comment · Reviewer_Q4am · 2025-03-14
> > >
> > > I would like to extend my gratitude to the authors for their comprehensive response. The addition of new experiments and model performance analysis has substantially enhanced the quality of the paper. Consequently, I am pleased to revise my rating from borderline to weak accept.

---

> > > > ### Author Response · Authors · 2025-03-14
> > > > **Official Reply to Reviewer Q4am**
> > > >
> > > > We sincerely appreciate your time and effort in reviewing our work and providing valuable feedback throughout the process. Your insightful comments and suggestions have greatly contributed to improving the quality and clarity of our manuscript. The addition of new experiments and model performance analysis was guided by your recommendations, and we are pleased that these refinements have enhanced the paper.
> > > >
> > > > We are grateful for your constructive engagement and thoughtful evaluation, which have strengthened our study. Your feedback has been invaluable in ensuring the robustness and rigor of our research, and we look forward to further developing this work in future studies.
> > > >
> > > > Thank you once again for your support and for helping us make this a stronger contribution to the field.

---

### Official Review · Reviewer_XMF3 · 2025-02-19

**Confidence:** 4
**Preliminary Rating:** 4
**Recommendation:** Poster
**Final Rating:** 4

**Summary:**

The paper introduces LUV-Net, a deep learning model designed for multi-label classification of lung ultrasound (LUS) videos, focusing on recognizing multiple LUS patterns such as A-lines, B-lines, consolidation, and pleural effusion. The key innovation lies in combining pattern-specific attention mechanisms with an efficient temporal feature extraction module, tailored to capture both spatial and temporal dependencies in LUS videos. The spatial module utilizes independent attention mechanisms for each LUS pattern, enhancing the model's ability to focus on relevant features specific to each pattern. The temporal module employs a 1D convolutional network to capture sequential relationships between adjacent frames efficiently. The model was evaluated on a development set of 341 LUS videos and a temporally separated validation set of 56 videos, demonstrating superior performance over conventional video models and achieving higher Area Under the Curve (AUC) scores across all patterns. The authors also validated the model's interpretability by visualizing pattern-specific attention regions, providing insights into the decision-making process. The code is publicly available, promoting reproducibility.

**Strengths:**

- The paper presents a combination of pattern-specific attention mechanisms with efficient temporal feature extraction, specifically tailored for multi-label classification in LUS videos.
- Unlike prior works focusing on single-pattern or disease-specific classifications, this model addresses the challenge of recognizing multiple coexisting LUS patterns simultaneously.
- The model is evaluated on both a development set and a temporally separated validation set, demonstrating consistent and performance across all LUS patterns compared to several baseline models.
- The visualization of pattern-specific attention scores enhances the interpretability of the model, allowing for better understanding and trust in the results by clinicians.
- The authors provide the code publicly, facilitating reproducibility and further research in the field.

**Weaknesses:**

- The datasets used for training and validation are collected from a single institution, which may limit the generalizability of the model to other clinical settings with different ultrasound machines or patient populations.
- While the model is compared with several baseline models, it lacks comparison with the latest state-of-the-art models in medical video analysis that could offer a more rigorous benchmark.
- The model's performance seems to heavily rely on specific hyperparameters (e.g., kernel size, number of attention heads), but there is limited discussion on how sensitive the model is to these choices in different scenarios.
- The use of DenseNet-161 as the encoder might introduce significant computational overhead, which could hinder real-time applications or deployment in resource-constrained environments.
- The paper lacks a reader study or clinical validation to assess how the model's predictions could impact clinical decision-making or improve patient outcomes in practice.

**Detailed Comments:**

- It would be beneficial to include more details about the patient demographics and pathologies represented in the datasets to assess the diversity and potential biases.
- Consider discussing plans for external validation on datasets from other institutions to strengthen the claims about the model's applicability in different clinical settings.
- Exploring lighter-weight encoders or model compression techniques could make the model more suitable for deployment on portable ultrasound devices.
- Providing a more detailed justification for the chosen hyperparameters and how they affect the model's performance could help others in reproducing and extending the work.
- Discuss how the model can be integrated into existing clinical workflows and any potential challenges that may arise.

**Justification Of The Final Rating:**

Thank you to the authors for the detailed response. I will maintain my rating at 4, as external validation and generalizability are designated as future work rather than being fully addressed in the current study, but I fully understand the challenges of tackling these aspects immediately.

**Justification Of The Preliminary Rating:**

The paper addresses a relevant and challenging problem in medical imaging by proposing a novel deep learning model, LUV-Net, for multi-label classification of LUS patterns. The approach is innovative, leveraging pattern-specific attention mechanisms and efficient temporal feature extraction, and demonstrates superior performance over baseline models. The inclusion of interpretability through attention visualization is a strength. However, the paper has some weaknesses, primarily concerning the generalizability due to the use of data from a single institution and the lack of comparison with the latest state-of-the-art models. Additionally, considerations regarding computational efficiency and real-world clinical validation are necessary.

**Questions To Address In The Rebuttal:**

- Can the authors provide additional results or discuss the model's performance on external datasets to assess its generalizability?
- How does the model compare with the latest state-of-the-art models in medical video analysis, not just conventional video models?
- Can the authors comment on the computational requirements and potential strategies to optimize the model for real-time applications?
- What measures have been taken to ensure the model does not introduce biases due to the limited diversity of the training data?
- Are there any plans to conduct a reader study or clinical validation to evaluate the practical impact of the model's predictions?

**Special Issue:**

No

---

> ### Author Response · Authors · 2025-03-07
> **Official Reply to Reviewer XMF3**
>
> We sincerely appreciate the reviewer's detailed and constructive feedback. Your insights have greatly helped us refine and improve our work. Below, we address each of the points raised.
>
> **1) Generalizability of the Model and External Datasets**
>
> We acknowledge the limitations of using a single-institution dataset. We explored publicly available lung ultrasound datasets, but most focus on specific patterns (e.g., COVID-19) or provide only image-level annotations rather than video-level data. To mitigate this, we employed a temporally separated validation set with expert annotations by clinicians with over eight years of experience.
>
>  For future work, we are currently in discussions with external hospitals to collect multi-institutional lung ultrasound data. Additionally, we plan to extend our model to support both image-level and video-level inputs, which will enable us to better leverage existing public datasets that primarily contain image-level annotations.
>
> **2) Comparison with State-of-the-Art Models**
>
> We appreciate the reviewer’s concern regarding comparisons with more recent state-of-the-art (SOTA) models.
> Our baseline selection was driven by the prevalent use of conventional architectures (R2Plus1D, C3D, CNN+LSTM) in LUS research, while direct comparison with LUS-specific models was challenging due to lack of publicly available code.
>
>  As suggested by the reviewer, we conducted additional experiments with ViT-based MViT and Video Swin Transformer from Torchvision. However, since differences in frame resolution and clip length prevented the use of pretrained weights, we trained these models from scratch under identical conditions.
> The results are included in Tables 1 and 2 of the revised manuscript.
>
> **3) Computational Efficiency and Model Compression**
>
> We appreciate the reviewer's comments regarding the computational cost of LUV-Net, particularly its applicability to portable ultrasound devices. Our study initially used DenseNet-161 as the image encoder, which, as the reviewer pointed out, introduces computational overhead.To address this concern, we conducted an additional experiment using the more lightweight ResNet-50 as the encoder, which maintained strong performance while reducing computational overhead, with results included in Table 4 of the revised paper.
>
>  As a solution, we plan to explore model compression techniques such as pruning and quantization to reduce computational cost. Additionally, we aim to optimize both the image encoder and attention mechanism in future work to enhance deployment feasibility.
>
> **4) Hyperparameter Justification**
>
> We also recognize the need for further discussion regarding hyperparameter selection and apologize for any lack of clarity in our initial submission.
>
>   * **Kernel Size of 1D Convolution:** The kernel size was chosen through a hyperparameter search ranging from 1 to 29, with the best validation performance observed at a kernel size of 13. A larger kernel size captures broader temporal dependencies, while a smaller size focuses on short-term relationships. However, providing an exact physiological rationale for selecting 13 is challenging, as LUS pattern duration varies significantly depending on patient conditions and Imaging acquisition protocol. Despite this, our empirical results indicate that intermediate value(13) effectively captures both short-duration artifacts and longer temporal dependencies present in LUS sequences.
>
>   * **Number of Attention Heads:** Similarly, We evaluated attention heads across values from 1 to 96, with 8 yielding the best validation performance. Increasing the number beyond this did not significantly improve performance, while reducing it led to a decline in feature expressiveness.
>
> **5) Clinical Validation and Reader Study**
>
> We agree with the reviewer that a reader study would significantly enhance the clinical applicability of our model. We are currently planning a study that focuses on key objectives. First, we aim to compare clinician performance with and without the model to assess its impact on LUS interpretation. Second, we will investigate whether real-time model predictions and key frame suggestions improve clinical decision-making. Lastly, we intend to evaluate the model’s usefulness across different clinician expertise levels to determine if it provides more substantial benefits for less experienced practitioners. These planned studies will help assess the model's practical value in real-world clinical settings.

---

### Official Review · Reviewer_Rvbx · 2025-02-21

**Confidence:** 5
**Preliminary Rating:** 4
**Recommendation:** Poster

**Summary:**

The manuscript proposes LUV-Net, a deep learning model designed for multi-label classification
of lung ultrasound (LUS) patterns, utilizing pattern-specific attention mechanisms and efficient
temporal feature extraction. The model is evaluated on both a development set and a temporally
separated validation set, showing strong performance in comparison to traditional video models.
The model's interpretability is enhanced through attention visualizations, and the work addresses
an important clinical problem in automating LUS analysis.

**Strengths:**

The code being made publicly available is a positive aspect, as it allows other
researchers to replicate the study and build upon it, enhancing the model's impact.

The incorporation of attention mechanisms is a key strength. By visualizing the
attention regions, the authors provide insights into the model’s decision-making
process. This is especially important in clinical settings, where model transparency
is crucial for gaining trust and facilitating adoption by medical professionals.

The experimental setup is comprehensive, including comparisons with state-of-the-
art models (C3D, R2Plus1D, CNN+LSTM, and USVN) and ablation studies that
test various aspects of the architecture, such as temporal modules, attention heads,
kernel size, and input frame lengths. These rigorous experiments demonstrate the
model's robustness and improve confidence in its results.

The use of statistical significance testing (p<0.05) further supports the validity of
the findings. The ablation study clearly shows that the inclusion of temporal
features leads to performance improvements, as evidenced by higher Macro-AUC
scores (e.g., Macro-AUC improves from 0.885 to 0.894).

The dual focus on temporal and spatial dependencies allows the model to
effectively handle the dynamic nature of ultrasound video sequences.

**Weaknesses:**

The primary weakness lies in the dataset size and the fact that both the development
and validation sets come from a single institution. This limits the generalizability
of the model, as ultrasound images can vary significantly across institutions in
terms of equipment, operator expertise, and patient demographics.
The small dataset size (341 training videos) also raises concerns about the model’s
robustness. Larger, more diverse datasets, including data from multiple institutions,
would enhance the model’s generalizability and reduce potential bias.

The use of four pattern-specific attention layers introduces additional parameters
and computational costs. The authors do not report the total number of learnable
parameters or compare the computational overhead (e.g., FLOPs, inference time)
against baseline models. Including this analysis would clarify whether the
performance gains justify the increased complexity.

The fixed kernel size (13) for 1D convolution is empirically justified but lacks
theoretical grounding. Discussing why local temporal dependencies are best
captured by this size (e.g., physiological rationale for frame relationships) would
add depth.

The paper acknowledges that the DenseNet-161 encoder is computationally
expensive, which may limit its applicability in real-time clinical scenarios,
especially in low-resource environments. Exploring lighter architectures or
optimization techniques could enhance the model's real-world usability.

While LUV-Net is compared with several existing models, there is no direct
comparison with other modern multi-label classification approaches, especially
those that specifically address medical image/video analysis. Including such
comparisons would provide a more complete context for evaluating the model’s
strengths and weaknesses.

The description of global query vectors (Section 2.3) is unclear. Specifically:
How are these vectors initialized? Are they shared across patterns or learned
independently?
What is the relationship between the global queries and the spatial features?
A mathematical formulation or diagram would improve clarity.

Table Citations: Tables 1 and 2 compare LUV-Net to models like USVN, C3D,
and R2Plus1D, but no references are provided for these baselines. Each model
should be explicitly cited (e.g., "C3D [25]") to ensure reproducibility and credit.

Parameter Efficiency: A table summarizing model sizes (parameters),
computational costs, and inference times for LUV-Net and baselines would
strengthen the technical evaluation.

**Detailed Comments:**

Expand the Dataset:
Incorporate data from multiple institutions to improve the generalizability and
robustness of the model. This would also help address concerns about dataset
biases.
Consider data augmentation strategies or synthetic data generation to further
augment the small dataset and enhance the model’s performance.

Enhance Real-World Applicability:Explore the use of lightweight model architectures or pruning techniques to reduce
computational costs, enabling real-time deployment in resource-constrained
settings.

**Justification Of The Preliminary Rating:**

The proposed LUV-Net model is a strong contribution to the field of medical imaging, particularly
in the automation of lung ultrasound analysis. It presents an innovative approach that combines
pattern-specific attention with temporal feature extraction, yielding high performance across
multiple metrics. However, the model’s applicability is limited by dataset size, generalizability
concerns, and computational cost. The paper would benefit from additional real-world validation,
error analysis, and technical clarifications. Despite these weaknesses, the manuscript demonstrates
significant potential and is highly relevant to medical imaging communities. I recommend
acceptance with minor revisions, particularly in expanding the dataset, improving model
efficiency for real-world use, and further clarifying key design decisions.

**Questions To Address In The Rebuttal:**

Improve Presentation:

The appendix contains important details, such as hyperparameters, that could be
summarized in the main body to improve the readability of the manuscript.

Some figures, particularly attention plots (Figure 3), are pixelated. Higher-
resolution visuals would enhance the clarity and interpretation of these critical
results.

Conduct real-world validation by testing the model in clinical environments. A
reader study, where clinicians evaluate the model’s predictions alongside their own,
would be crucial in assessing its practical utility.

The baselines (e.g., C3D, CNN+LSTM) are well-established but outdated.
Comparisons with recent LUS-specific models are missing. The authors should
justify their choice of baselines and include state-of-the-art alternatives.

**Special Issue:**

No

---

> ### Author Response · Authors · 2025-03-07
> **Official Reply to Reviewer Rvbx**
>
> We sincerely appreciate the detailed and constructive feedback provided by the reviewer. The comments have significantly helped us refine and improve our work. Below, we address each of the points raised.
>
> **1) Limitations of Dataset Generalizability & Future Work**
>
> We acknowledge the limitations of using a single-institution dataset. While publicly available lung ultrasound datasets exist, they primarily focus on COVID-19 cases or provide only image-level data, making them unsuitable for our video-based approach.
>
> To improve generalizability, we designed a temporally separated validation set and ensured expert annotations by clinicians with over eight years of experience. However, as the reviewer correctly pointed out, variability in ultrasound images across institutions, operators, and patient demographics remains a key challenge. Future work will incorporate multi-institutional data and diverse ultrasound vendors to address this limitation.
>
> **2) Kernel Size Justification**
>
> We agree that understanding the physiological rationale for frame relationships is crucial.
> However, discussions with clinicians highlighted that variations in patient conditions and the routine imaging acquisition process introduce complexities that make it difficult to define a universally optimal kernel size.
> Notably, LUS patterns can exhibit both short and extended temporal variations depending on the patient's condition(disease severity) and the routine imaging acquisition process.
>
> To accommodate this variability, we conducted a hyperparameter search across values from 1 to 29, selecting the configuration that achieved the best validation performance (Table 8).
> The model performed best at a kernel size of 13, suggesting that this intermediate value effectively captures both short-duration artifacts and longer temporal dependencies present in LUS sequences.
>
> This result implies that the optimal kernel size must accommodate the variability in LUS pattern duration, effectively capturing both shorter and longer pattern durations, which explains why an intermediate value performed best.
>
> **3) Model Efficiency Metrics**
>
> We appreciate the reviewer’s comments regarding the computational efficiency of LUV-Net. To further clarify this aspect, we have added a comparison of model parameters, GFLOPs, and inference time in the revised manuscript. Additionally, to explore a more efficient variant of LUV-Net, we conducted further experiments using ResNet-50 as the encoder instead of DenseNet-161(Table 3).
>
> **4) Query Vector Initialization**
>
> We appreciate the reviewer's question regarding the initialization and role of query vectors in our model. Below, we clarify key aspects:
>
>   * **Query vectors Initialization** : The query vectors are initialized using a normal distribution and are learned during training. Specifically, they are defined as trainable parameters, initialized with small random values, and progressively optimized as the model learns attention weights.
>   * **Independence Across Labels**: Each query vector is independently learned for each label. That is, the model maintains a distinct query vector per label, ensuring that attention computations remain specialized for each classification task.
>   * **relationship between global query vectors and spatial features**: The query vectors interact with the feature map through dot-product operations, followed by softmax normalization, which generates attention scores. These scores guide the model to extract label-specific information from the input frames, helping to differentiate spatial features for each class effectively.
>
> **5) Baseline Model Citations**
>
> We apologize for the omission of citations in the previous version.
> We have now included proper references for all baseline models (e.g., C3D, R2Plus1D, and USVN) to ensure reproducibility and proper attribution.
>
> **6)Visualization Quality**
>
> We apologize for the low resolution of Figure 3. In the revised edition, we have improved the attention plot resolution to enhance readability and interpretation.
>
> **7) Comparison with State-of-the-Art Models**
>
> We acknowledge the importance of comparing our model with recent LUS-specific and state-of-the-art video models.
> Our baseline selection was primarily driven by the prevalent use of R2Plus1D, C3D, and CNN+LSTM architectures in most prior LUS studies, while direct comparison with LUS-specific models was challenging due to the absence of publicly available code and weights.
>
> As suggested by the reviewer, we expanded our evaluation by conducting additional experiments with ViT-based architectures (MViT-B and Swin-B from torchvision). However, Due to differences in frame resolution and clip length compared to their original pretraining datasets, we could not use pretrained weights. Instead, we trained them from scratch under identical conditions and evaluated their performance accordingly. (Table 1, 2).

---

### Author Rebuttal · Authors · 2025-03-07

**Rebuttal:**

We sincerely appreciate your detailed feedback, which has significantly improved our work. Below, we summarize our key revisions and clarifications.

**1) Dataset Generalizability**

We acknowledge the limitation of using a single-institution dataset. While publicly available LUS datasets exist, most focus on specific conditions (e.g., COVID-19) or contain only image-level annotations. To address this, we used a temporally separated validation set and expert annotations. Additionally, we are collaborating with external hospitals to collect multi-institutional LUS data. Furthermore, to leverage existing public datasets, we are enhancing our model to support both image- and video-based inputs, allowing for more flexible training and evaluation.

**2) Kernel Size Justification**

LUS patterns vary in duration, making a fixed kernel size challenging. We conducted a hyperparameter search (1–29) and selected 13, which effectively captures both short-duration artifacts and long-range dependencies.

**3) Model Efficiency**

We expanded our analysis by adding GFLOPs, parameter, and inference time comparisons. We also replaced DenseNet-161 with ResNet-50, maintaining strong performance while reducing computational cost (Table 3). Future work will explore pruning and quantization to optimize real-time deployment.

**4) R2Plus1D vs. C3D Performance**

Despite R2Plus1D's success in action recognition, C3D outperformed it in LUS classification due to key differences:

- LUS patterns are localized, whereas action recognition requires long-range motion modeling.
- C3D’s 3D convolutions capture spatial-temporal dependencies, while R2Plus1D’s 2D+1D approach focuses on long-term motion, making it less suited for LUS.
- These results suggest C3D's architecture is better suited for LUS classification.

**5) Query Vector Learning & Attention**

Each query vector is independently learned per label, ensuring class-specific attention. We clarified how query vectors interact with spatial features via dot-product attention.

**6) Transformer-Based Models**

Experiments with MViT-B and Swin-B showed suboptimal performance due to differences in frame resolution and clip length from their pretraining datasets, requiring training from scratch (Tables 1 & 2). This highlights the need for LUS-specific pretraining in future work.

**7) Visualization Improvements**

We improved the resolution of Figure 3 and plan to analyze misclassified cases to enhance interpretability.

**Supporting Material:**

/attachment/4c18d5cbb85b8a6c9cc747f73eed0603b1f6710d.pdf

---

### Comment · Area_Chair_xRwa · 2025-03-07
**Paper is open for disucssions**

The authors have provided their answers to the raised comments. They have also provided a revised version with changes highlighted. Reviewers are encouraged to read both and update their initial review.

---

### Meta-Review · Area_Chair_xRwa · 2025-03-18

**Recommendation:** Accept (Poster)
**Confidence:** 5

**Metareview:**

The authors have effectively addressed the main concerns raised by the reviewers in their rebuttal, which led to a unanimous decision of weak acceptance. While the evaluation on multi-institutional datasets remains an open question, the overall quality and contributions of the work warrant its acceptance for presentation at the 2025 MIDL meeting.